



# Towards a regional high-resolution bathymetry of the North West Shelf of Australia based on Sentinel-2 satellite images, 3D seismic surveys and historical datasets.

Ulysse Lebrec[1, 2*], Victorien Paumard[1], Michael J. O'Leary[1] and Simon C. Lang[1]

[1]Centre for Energy Geoscience, School of Earth Sciences, The University of Western Australia, 35 Stirling Highway, Perth, WA, 6009, Australia
[2]Norwegian Geotechnical Institute, 40 St Georges Terrace, Perth, WA, 6000, Australia

*  *Correspondence to*: Ulysse Lebrec (ulysse.lebrec@research.uwa.edu.au)

## Abstract

High-resolution bathymetry is a critical dataset for marine geoscientists. It can be used to characterize the seafloor and its marine habitats, to understand past sedimentary records and even to support the development of offshore engineering projects. Most methods to acquire bathymetry data are costly and can only be practically deployed on relatively small areas. It is therefore critical to develop cost-effective and advanced techniques to produce large-scale bathymetry datasets.

This paper presents an integrated workflow that builds on satellites images and 3D seismic surveys, integrated with historical depth soundings, to generate a regional high-resolution digital elevation model. The method was applied to the southern half of Australia's North West Shelf and led to the creation of a new high-resolution bathymetry, with a resolution of 10 x 10 m in nearshore areas and 30 x 30 m elsewhere.

The vertical and spatial accuracy of the datasets have been thoroughly assessed using open source Laser Airborne Depth Sounder (LADS) and Multi Beam Echo Sounder (MBES) surveys as a reference. The comparison of the datasets indicates that the seismic-derived bathymetry has a vertical accuracy better than 1 m + 2% of the absolute water depths, while the satellite-derived bathymetry has a depth accuracy better than 1 m + 5% of the absolute water depths. This dataset constitutes a significant improvement of the pre-existing regional 250 x 250 m grid and will support the onset of research projects on costal morphologies, marine habitats, archaeology, and sedimentology.

All datasets used as inputs are publicly available and the method is fully integrated in Python scripts making it readily applicable elsewhere in Australia and around the world. The workflow as well as the resulting bathymetry have been independently reviewed and approved for release by a technical committee from the AusSeabed Community (Geoscience Australia). The regional digital elevation model as well as the underlying datasets can be accessed at: https://doi.org/10.26186/144600.


# 1    Introduction

The North West Shelf (NWS) is a ~2,400 km long carbonate platform spanning along the northern margin of Australia, between
10° S and 25° S (James et al., 2004). The shelf is composed of two parts located on either side of longitude 123°E (Fig. 1).
The Rowley Shelf extend westward to Exmouth while the Sahul Shelf extends eastward to the Melville Island (Fairbridge,
1953). The NWS region, which commonly includes the adjacent plateaus and terraces, is a hotspot of biodiversity and hosts
several marine conservation parks (Wilson, 2013; Australian-Marine-Parks, 2018). The NWS is also a site of key

archaeological significance as it may have been one of the entry points for humans into Australia (Veth, 2017). Finally the
region is Australia's main hydrocarbon province (Purcell and Purcell, 1988) and concentrates significant fishing activities
(Nowara, 2001). Nevertheless, most of the shelf seafloor, marine habitats and biodiversity remain poorly understood (James
et al., 2004; Wilson, 2013; Poore et al., 2015).

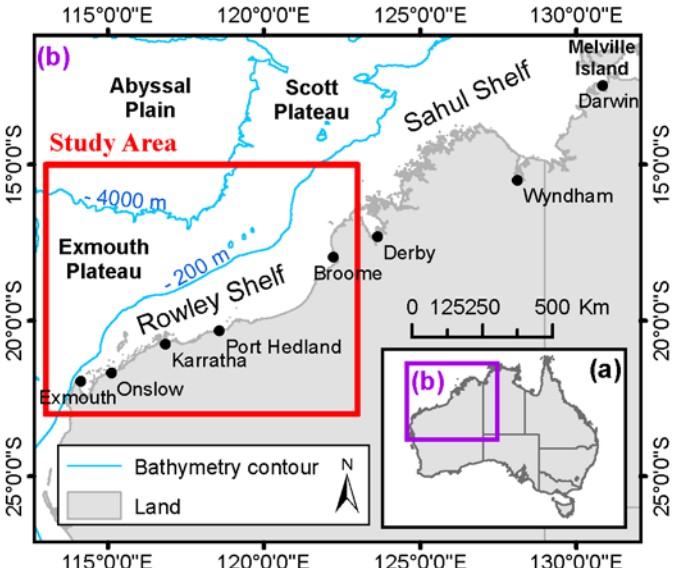

**Figure 1: Location of the study area. The area of interest engulfs the Rowley Shelf (Southern half of the North West Shelf) and the
adjacent plateaus**

Investigations into the modern depositional environments of the NWS typically cover relatively small areas. Some regional
studies exist, but they almost solely build on sparse seabed samples (Carrigy and Fairbridge, 1954; Jones, 1973; Dix, 1989;
James et al., 2004; Baker et al., 2008). This observation can be explained by the limited coverage of open source high-resolution

geophysical datasets (Fig. 2), which in turn can be explained by the prohibitive cost of acquiring such datasets.
Geoscience Australia and the AusSeabed community are leading the effort to map the seabed around Australia via the
acquisition and distribution of Multi Beam Echo Sounder (MBES) bathymetry (Spinoccia, 2018). As of now, less than 25%
of the Australian waters, have been mapped using high-resolution techniques (Picard et al., 2018). Along the North West Shelf,
this number drops below 15%. The integration of low-resolution and indirect datasets can however allow the interpolation of

high-resolution grids and therefore help reduce the extent of poorly charted areas. Based on this approach, high-resolution



bathymetry compilations were created over the Sahul Shelf, the Northern Territory and the Great Barrier Reef using MBES data, water-depth measurements from navigation charts, LiDAR surveys, satellites data and single beam surveys (Beaman, 2017, 2018).

**Figure 2: Public data available within the study area. ENC navigation charts and the Australian Topography and Bathymetry grid cover the entire area.**

The intent of this research was to follow a similar approach to create the first high-resolution bathymetry compilation across the Rowley shelf and in doing so complete the existing work conducted over the Sahul shelf to provide a full bathymetric coverage over the NWS. The main challenge to generate such compilation is that the underlying high-resolution bathymetry surveys have a limited coverage and most of the areas consist of resampled or interpolated low-resolution datasets. To work around this limitation, we developed an innovative workflow to derive high-resolution bathymetry from satellite images and

3D seismic surveys at a regional scale. The workflow was successfully applied to the area of interest and the resulting product was integrated with pre-existing open source datasets to create a regional 30 x 30 m Digital Elevation Model (DEM) over an area of 935,000 km$^2$.

The scientific objective of this paper is to introduce the bathymetry compilation produced over the Rowley Shelf and the associated workflows to produce this outcome. The dataset includes the derived digital elevation model as well as the underlying datasets in their original resolution. In the following sections, we present the data selection, the workflows used to produce the satellite-derived bathymetry and the seismic-derived bathymetry, as well as the processes adopted to compile a seamless high-resolution bathymetry. Quality check processes and discussion on the vertical and position accuracy are

presented for each dataset included in the compilation.

## 2    Processing tools

Datasets presented in this paper were processed using the Python programming language. Python scripts were developed using three key libraries: (1) raster and shapefile calculations build on ArcGIS geo-processing tools, accessed via the Arcpy library; (2) scikit-learn library, an open source machine learning Python library was used to perform statistical analysis of the data;

and (3) the required computations were dynamically split between the logical cores of the workstation using Python multiprocessing module. While the workflows were fully integrated in scripts, each of the processing step presented in this paper can be completed manually via ArcGIS, and presumably using any other GIS software. Software used for specific processing steps are presented in the relevant sub-sections.

## 3    Pre-existing datasets

### 3.1    Australian bathymetry and topography grid

This dataset was released by Geoscience Australia in 2009 and covers the totality of the Australian waters and mainland with a bin size of 9 arc seconds (1 arc second ≈ 30 m near the equator). The bathymetry was generated via the integration of direct water-depth soundings acquired or collated by Geoscience Australia with 1 arc minute ETOPO and 2 arc minute ETOPO bathymetry (Whiteway, 2009). Within the area of interest (Fig. 2), the topography is based on the Australian GEODATA 9

second Digital Elevation Model (DEM) (Whiteway, 2009). The bin size does not always reflect the resolution of the data as extended areas, especially in deep-water domains, were interpolated from sparse data points.

### 3.2    SRTM-derived digital elevation model

Geoscience Australia produced the ground surface topography using SRTM data acquired by the NASA in February 2000 (Gallant, 2011). The data was produced by the authors using an automated process to remove the vegetation from the original

SRTM data and was released in 2011 as a 1 arc second grid.



### 3.3 Multi Beam Echo Sounder bathymetry

Multi Beam Echo Sounder bathymetry acquired around Australia by numerous institutions have been collated and merged by Geoscience Australia (Spinoccia, 2018). The data pack, which covers limited areas (Fig. 2), is regularly updated and is available as a 50 m grid. The distance between the recorded points of a beam increases with the water depth and vice versa (Lurton, 2002), hence the average bin size of 50 m used by Geoscience Australia does not always correspond to the actual resolution of the data (i.e., the points can be spaced by only a few centimetres in the shallowest waters). To overcome this issue, the native xyz data was obtained from Geoscience Australia and re-gridded using the root square of the average point density of each survey line as bin size. In some instances, where xyz files were not available, the re-gridded datasets were complemented with the 50 m grid.

### 3.4 National intertidal digital elevation model

This digital elevation model was compiled by Bishop-Taylor et al. (2019) using the Inter-Tidal Extent Model (ITEM) developed by (Sagar et al., 2018; Sagar et al., 2017) and integrates 30 years of Landsat images. The grid has a resolution of 25 x 25 m and covers intertidal areas (Fig. 2) typically comprised between +5 m and -5 m Mean Sea Level (MSL).

### 3.5 ENC Navigation chart

Navigation charts virtually cover the whole area of interest and were sourced from the Australian Hydrographic Office. Depth soundings extracted from the charts constitute a unique source of verified water-depth measurements and can be used to calibrate other surveys. Point's density varies significantly depending on the distance from the coast and proximity to highly navigable and/or populated coastal areas. On average, the distance between two measurements is comprised between 500 and 5000 m. Unlike other datasets that are reduced to the Mean Sea Level (MSL), the navigation charts are referenced with respect to the Lowest Astronomical Tide (LAT). To ensure the proper use of this dataset, all depth soundings were converted from LAT to MSL using the Australian Coastal Vertical Datum Transformation Software (AUSCoastVDT), provided by the Intergovernmental Committee on Surveying and Mapping (ICSM). The software allows the conversion between vertical datum down to a water depth of approximately 500 m (Crcsi and Frontiersi, 2019). Beyond that depth, the vertical uncertainty related to the datum is considered below the vertical accuracy of the depth sensors and hence of negligible impact.

### 3.6 Open source LADS

Laser Airborne Depth Sounder (LADS) data was collected from the West Australian Government's Department of Transport. The data was acquired by Fugro in the vicinity of Onslow and Barrow Island. Surveys were conducted between 1998 and 2002 and covers area of ~4200 km$^2$. The spatial resolution of the data varies but is typically < 5 m.



## 4    Seismic-derived bathymetry

### 4.1    Overview

Australia's North West Shelf has been extensively surveyed, with the shelf between Exmouth and Broome now covered by ~325,000 km$^2$ of 3D seismic survey (Paumard et al., 2019b). Under Australian legislation, most of this extensive dataset is publicly available through Geoscience Australia. The bathymetry can be derived in two ways, either by extracting the first seismic reflection from the data itself (which typically represents the seafloor) or by compiling the water-depth measurements

from the vessel echo sounder.

### 4.2    Reflection-derived bathymetry

#### 4.2.1    Data source

Open-file seismic data were provided by Geoscience Australia. In total, 26 publicly available seismic surveys were processed (Fig. 3). Additional surveys are integrated as they become available.

#### 4.2.2    Data processing

The extraction of the seabed reflection was performed using PaleoScan™, a full volume seismic interpretation software. The interpretation is performed semi-automatically using similarities between adjacent seismic traces to generate an unlimited number of horizons within a chronostratigraphic framework (Paumard et al., 2019a). In this case, the workflow focused on the upper 500 ms (TWT) of each seismic survey to optimize the resolution of the seafloor horizon.

Seismic horizons were subsequently converted from the time domain to depth domain. Due to the lack of regional water-column velocity profile, an average velocity of 1500 m/s value was obtained by averaging the nominal velocities specified in the navigation files of the surveys. The resulting velocity, while averaged from indirect sources, is comparable with values used in the literature to perform similar conversions (Mosher et al., 2006; Jibrin et al., 2013; Power and Clarke, 2019). The bin size of the reflection-derived bathymetry grids corresponds to the spacing of the seismic traces and is generally comprised

between 12.5 m and 37.5 m.





**Figure 3 Extent of the reflection-derived and navigation-derived bathymetry. All areas covered by the reflection datasets are also covered by the navigation datasets.**

### 4.2.3    Data limitation

The geometry of multichannel seismic survey acquisition systems appears to result in a reduction of the vertical resolution in water depth of less than 150 m. As this depth reduces, the bathymetry becomes increasingly noisy and the morphologies start to be vertically distorted, to the point where the relative height of seabed features can be multiplied by a factor 5 compared to MBES data (Fig. 4, b). Similar patterns were identified by the U.S. Bureau of Ocean Energy Management in the seismic-derived bathymetry they generated over the Gulf of Mexico (Kramer, 2017). In addition, sound velocity varies in the water

column depending on the salinity, temperature and pressure of the seawater (Leroy, 1969). Therefore, the use of a constant

velocity to perform the time-depth conversion cannot capture local variations of the velocity profile and may result in a local underestimation or overestimation of the water depths.

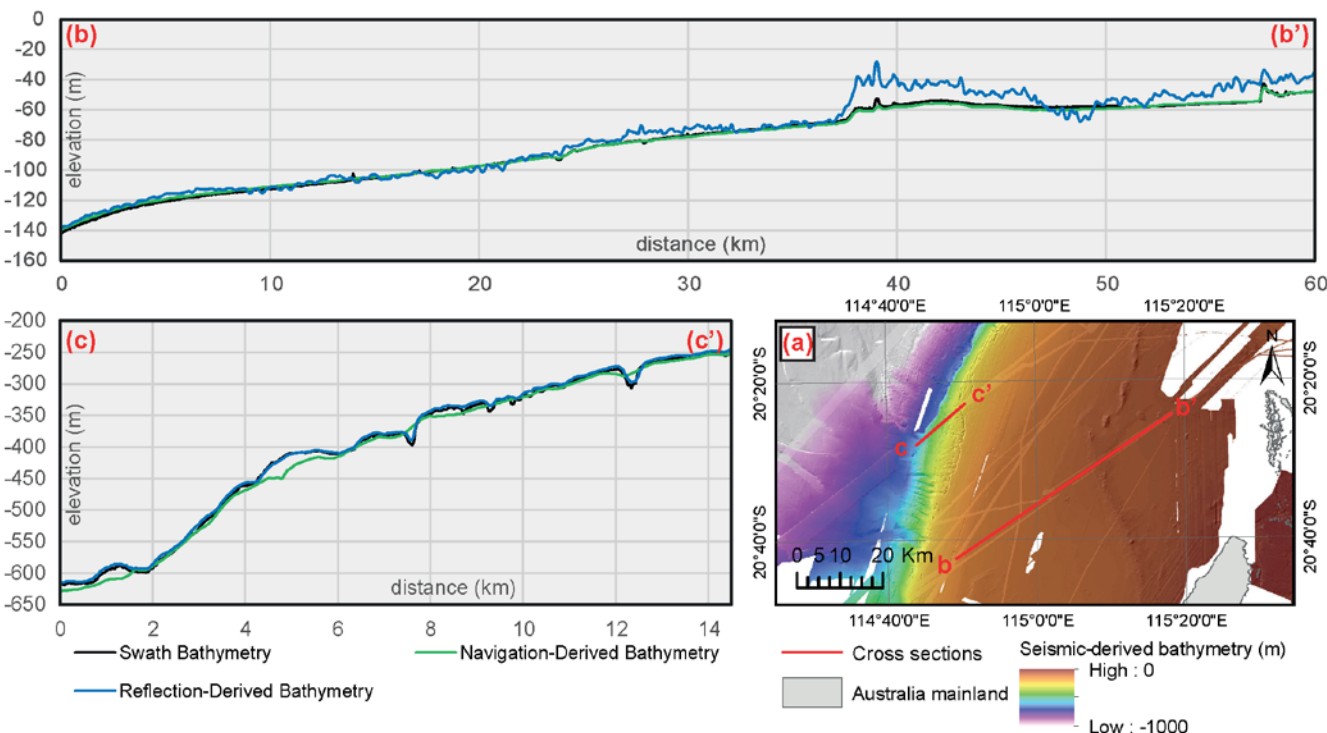

**Figure 4 Comparison of the navigation-derived and reflection-derived bathymetry with MBES surveys in the vicinity of Barrow Island (a). Reflection data is heavily distorted in shallow waters (cross-section b, from km 37) whereas navigation data is over-smoothed in deep waters (cross-section c, until km 10).**

## 4.3 Navigation-derived bathymetry

### 4.3.1 Data source

P1/90 files are generated during the seismic acquisition and contain navigation-related information which include, for example, the coordinates of the vessel, source, receivers, and subsequent common mid points. Each entry is associated to a water-depth measurement from the on-board echo sounder (Iogp, 1990). These files are, in most cases, publicly available making them a powerful input for the generation of bathymetry at a regional scale.

Geoscience Australia has previously undertaken the compilation of the water-depth measurements from both 3D seismic and 2D seismic navigation files into a database. This product was made available to the project but did not include navigation data from the latest 3D seismic surveys. The navigation files from the missing surveys were subsequently sourced from the National Offshore Petroleum Information Management System (NOPIMS) data portal. In total, 232 navigation files were collected from 3D seismic surveys within the area of interest (Fig. 3).





### 4.3.2    Data processing

During seismic acquisition, whenever a shot point is acquired, the coordinates of the different part of the acquisition system

are recorded in the navigation files and are associated to the water-depth value recorded by the echo sounder (Iogp, 1990).
Some parts of the acquisition system can be hundreds of meters from the actual echo sounder location, leading to the creation
of a significant mismatch between the actual location of the water-depth measurement and their coordinates (Fig. 5a). In several
instances, this information is not properly recorded in the file (e.g., either the field is empty or the coordinates from the echo
sounder are not recorded, or at the wrong location). The P1/90 files were resultantly filtered in an iterative process to check if

the water depth were properly recorded at the echo sounder location and if not, to use the values from the closest recorded
point on the vessel. Doing so significantly reduced the spatial offset between the acquisition lines (Fig. 5b).

Water-depth measurements from the navigation files are then interpolated and gridded using ArcGIS inverse distance weight
algorithm. Hengl (2006) suggests that the ideal bin size to interpolate a scattered cloud of point is the average minimum
distance between 2 points, divided by 2. In this case, this formula often provided inaccurate results due to the geometry of the

modern marine seismic surveys: data points are typically recorded every 6.25 m to 25 m along acquisition lines that are spaced
by multiple times this value (Vermeer, 2012). The minimum distance between two points is consequently not representative
of the overall point density. The formula was adapted to instead use the root-square of the average area occupied by a point,
divided by two. Navigation-derived bathymetry grids typically have a bin size comprised between 30 m and 50 m.

### 4.3.3    Data Limitation

This geometry of modern marine seismic surveys was taken into account in the definition of the ideal bin size but the gap
between the acquisition lines was sometimes too important compared to the point spacing along these lines to result in an
accurate bathymetry grid. In such instances, interpolation artefacts marking the acquisition lines can be seen on the grids.
Overall, the navigation-derived bathymetry tends to become smoother as the depth increase (Fig. 4, c).

It is also often unclear from the navigation files what level of correction was applied to the water-depth measurements. In some

cases, it was stipulated whether the recordings were corrected for the roll, tides, and waves. However, in most cases, this
information was not available making it impossible to have a homogeneous correction process for all surveys.

Finally, the headers of some of the files suggest that some measurements may have been converted from the time to depth
domain using constant value instead of site-specific velocity profiles resulting in a possible over-estimation or under-estimation
of the water depths.





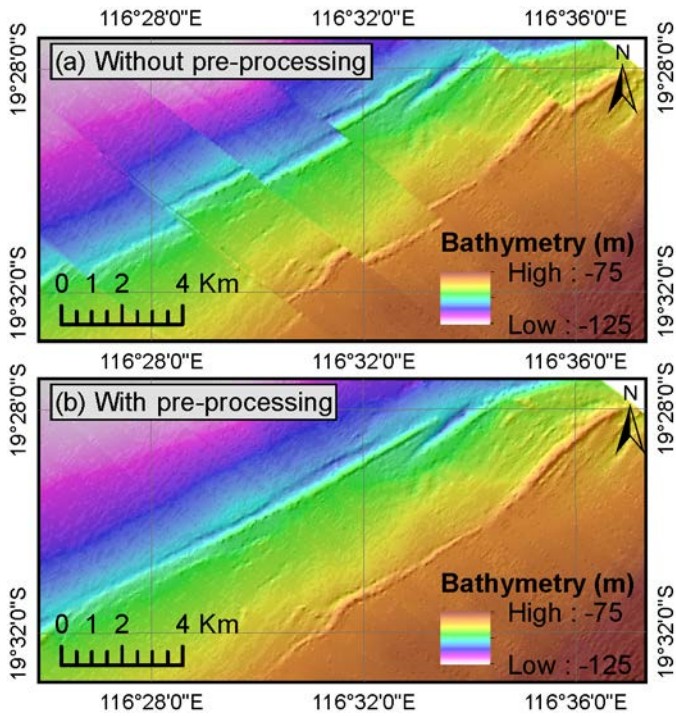


**Figure 5: Comparison of the navigation-derived bathymetry generated using the raw navigation files (a) or filtered navigation files (b) from Demeter 3D survey. In raw navigation files, water depth measurements are sometimes associated with the coordinates of the seismic receivers which can be located hundreds of meters behind the vessel hence creating a 'banded" pattern. During the filtering process, water-depth measurements are re-aligned with the vessel position.**

**4.4     Data calibration**

As the seismic data is archived in time (ms), the reflection-derived bathymetry lacks a consistent vertical reference system and a vertical offset of several meters can often be observed between the produced bathymetry files. Likewise, even though navigation files are supposedly reduced to the mean sea level, it is common to observe similar vertical offsets between surveys. The initial intent was to reduce all datasets to the mean sea level using the depth soundings from the navigation charts as

calibration points. However, in most cases, this method presented two limits: (1) the point density is often very low in deep waters, meaning that less than 10 points will generally intersect a given 3D seismic survey; and (2) the validity of the ENC depth soundings is sometimes questionable as vertical offsets of tens to hundreds of meters were locally observed with the MBES bathymetry.

Depth soundings from ENC tiles were subsequently complemented with depth values from the MBES bathymetry surveys.

MBES surveys having a limited coverage, many areas were not intersected by neither the ENC data points nor the MBES surveys. The reflection-derived and navigation-derived bathymetry were therefore vertically shifted in an iterative process to obtain the best fit between the ENC depth soundings, the MBES bathymetry and the surrounding seismic surveys.



## 4.5 Seismic-derived bathymetry compilation

Reflection-derived and navigation-derived bathymetry grids were then merged in order to generate a consistent, seamless

seismic-derived bathymetry (Fig. 6, a). In several areas, multiple surveys are overlapping. In such instances, due to variability of resolution between surveys, only the value from the most reliable survey was selected for inclusion in the final merge. Where appropriate, bathymetry grids were cropped to only include parts of them. The selection was performed considering two criteria. Firstly, the accuracy of the reflection-derived bathymetry varies depending on the water depth. Thus, where water depths are less than 150 m, the navigation-derived bathymetry is preferred, whereas the reflection-derived bathymetry is

preferred where the water depths are above 150 m. Secondly, surveys with the smallest line spacing and bin size (i.e., high-resolution surveys) were prioritized. Prior to their inclusion, individual grids were resampled using a bilinear interpolation and a bin size of 30 m.

**Figure 6: The seismic-derived bathymetry (a) is mainly composed of navigation-derived bathymetry (b') in water depth of less than**

**150 m and of reflection-derived bathymetry (c') in deeper water. 12,676 data points were extracted along a mesh from both MBES and seismic-derived bathymetry to assess the accuracy of the latest (d). The produced bathymetry marks a significant improvement compared to the Australian Bathymetry and Topography (b vs b' and c vs c').**



## 4.6 Data accuracy

The 30 m bathymetry grid resulting from the compilation of the reflection-derived bathymetry and navigation-derived bathymetry was compared with the MBES bathymetry publicly available on the North West Shelf to assess its vertical accuracy (Fig. 6, b, c, d). Values from both datasets were extracted along a 1000 m mesh, representing a total of 12,676 points, and were plotted against each other. The results show a very tight correlation between both datasets, hence confirming the quality of the final product. The Mean Absolute Error (MAE) gradually increases with the water depth but its relative percentage of the absolute water depth decreases, meaning that the vertical accuracy of the data increases with depth (Table 1).

**Table 1 Statistical metrics of the seismic-derived bathymetry**

| Water Depth interval (WD) in m | 20 -100 | 100 - 500 | 500 - 1000 | 1000 - 2000 | 2000 - 4000 |
|---|---|---|---|---|---|
| Mean Absolute error (MAE) | 1.18 | 4.37 | 6.87 | 7.2 | 14.84 |
| Root mean square error (RMSE) | 1.59 | 7.13 | 11.54 | 11.77 | 22.9 |
| MAE % of median depth | 2 %d | 1. 45 %d | 0.9 %d | 0.48 %d | 0.49 %d |

## 5 Satellite-derived bathymetry

### 5.1 Overview

The utilization of satellite images to derive bathymetry has been the focus of numerous papers since the late 1970s and relies on the idea that since the light's wavelengths are not absorbed homogeneously by the water, it should be possible to derive the 240 water depths from the spectral content of the light reflected by the seabed. Out of the various methods developed, two main approaches emerged: (1) the physical approach, which attempts to model the penetration of the light in the water and the resulting spectral content to estimate the bathymetry (Lee et al., 1994; Stephane et al., 1994; Brando et al., 2009); and (2) the empirical approach, which tries to correlate calibration points (i.e., true water-depth measurements) with the seabed reflectance (Lyzenga, 1978; Stumpf et al., 2003). Due to the extent of the area of interest and the high density of potential calibration 245 points available (e.g., navigation charts depth soundings), the empirical method has been preferred over the physical approach for this study.

The GEBCO cook book (Iho, 2018), published by the International Hydrographic Organization (IHO) and the Intergovernmental Oceanographic Commission (IOC) presents in great lengths a workflow to implement an empirical method using ArcGIS software. The method, which uses Landsat satellite images and calibration points from public navigation charts, 250 builds on the log-ratio equation (Eq. (1)) developed by Stumpf et al. (2003). The model is, according to the publication, less susceptible to sea bottom effect than the Lyzenga model.

$$z = m_1 \left( \frac{Ln(Blue)}{Ln(Green)} \right) - m_0 \qquad (1)$$

Where $m_1$ and $m_0$ correspond to empirically determined gain and offset and Blue and Green to the observed radiance.



The steps are as follow:

1.      Manual identification of the water/land separation and removal of the land using infra-red ("IR") band;

        2.      Generation of the band ratio using the Green and Blue bands;

        3.      Extraction of the band ratio values at calibration point locations;

        4.      Calculation of the average band ratio value per true water-depth measurement;

        5.      Visual definition of the depth of extinction (i.e., maximum depth of validity of the data);

6.      Generation of a linear regression line between the average band ratio values and the true water-depth measurement;

        7.      Application of the gain and offset of the regression line to the band ratio.

Following the guidelines from the IHO (Iho, 2018), the GEBCO workflow was used as a starting point in the development of the satellites-derived bathymetry.

The main limitation of this workflow is that the gain and the offset are calculated at the scale of satellites image tiles (tens to hundreds of km wide, depending on the satellite images used as input). These parameters therefore fail to capture spatial variation of seabed reflectance related to local environmental factors such as the type of substrate, the presence or absence of benthic communities, the presence or absence of marine plants.

Furthermore, existing studies following a GEBCO like workflow typically perform the processing steps manually and only
apply the workflow to unique satellite images for unique locations (Pe'eri et al., 2014; Hamylton et al., 2015; Kabiri, 2017; Casal et al., 2018; Caballero and Stumpf, 2019).The result is strongly impacted by temporal events affecting either the sea surface or the water column and is therefore difficult to reproduce.

Other authors have attempted to circumvent such temporal artefacts by using multiple images acquired through a given period of time over a specific area. While heading in the right directions, these studies usually include a limited number of image
(Chu et al., 2019; Evagorou et al., 2019; Poursanidis et al., 2019).

The number of manual steps required in the GEBCO workflow, as well the spatial and temporal uncertainties inherent to the empirical method, make the processing of satellite-derived bathymetry over large areas challenging and of extremely variable accuracy.

In order to allow the production of reproducible seamless satellite-derived bathymetry at a regional scale, the workflow has
been fully automated in Python scripts and was complemented by four processing steps, which aim at:

        1.      Improving and automating the water/land delineation using the Normalized Difference Water Index (NDWI) from Mcfeeters (1996) (Eq.(2));

        2.      Improving and automating the calculation of the depth of extinction;

        3.      Correcting the data for spatial variation of the seabed reflectance using regional residual error models; and

4.      Removing the effect of temporal events by the generation of statistics model using multi-temporal images for each pixel.



$$NDWI = \frac{Green - NIR}{Green + NIR} \tag{2}$$

Where Green corresponds to Sentinel band B02 and NIR to Sentinel band B08

## 5.2    Data Selection

### 5.2.1    Satellite data

The Sentinel-2 constellation is composed of two satellites launched by the European Space Agency (ESA) through the Copernicus program in 2015 and 2017 to acquire high-resolution (10 m to 60 m) multi-spectral images of the earth surface (Esa, 2020). The data acquired by the constellation, which have a position accuracy of 20 m on the ground, correspond to the highest resolution product publicly available and was therefore selected for this study.

Sentinel satellites images are available as tiles, each covering an area of approximately 100 x 100 km. In total, the area of interest is covered by 26 tiles. Images are composed of 13 spectral bands that are initially available as top-of-atmosphere reflectance products (Level 1c products). These images are not directly usable to derive the bathymetry as 90% of the reflectance actually correspond to the atmosphere (Gordon, 1983; Ioccg, 2010). In December 2018. Copernicus started to release Bottom-of-atmosphere products (Level-2a products) which are corrected for atmospheric effects using the Sens2cor tool (Gatti, 2018). In April 2020, Sentinel-Hub, a third-party company specialized in Earth observation, used the same tool to process from Level-1c to Level-2a all Sentinel-2 images acquired prior to December 2018 and decided to make them freely available (Milcinski, 2020). Thus, the complete catalogue of Sentinel 2 images is now fully and freely available as Level 2a products and can be used for further processing.

The penetration of the light in the water, and hence the accuracy and the penetration of the bathymetry produced, is highly dependent on seasonal environmental factors such as the cloud cover, turbidity of the water or roughness of the sea (Caballero and Stumpf, 2020; Zheng and Digiacomo, 2017). It is therefore necessary to select an optimum time window to minimize abnormal values included in the model.

Climate data available from the Bureau of Meteorology (Bom, 2020) suggest that the optimum environmental conditions (i.e., low precipitation and wind speed) may occur between August and November each year. This timeframe was further reduced to the period ranging from August to October following a visual inspection of the data. A total of 1,170 Level-2a Sentinel-2 images acquired in August, September, and October 2017, 2018 and 2019, with a cloud cover of less than 1%, were selected to be included in the bathymetry model.

### 5.2.2    Calibration points

The determination of the parameters m0 and m1 from the Stumpf et al. (2003) equation requires true water-depth measurements to use as calibration points. The depth soundings extracted from the navigation charts represents the main source of widely spread water-depth measurements on the North West Shelf. In total, more than 125,000 points are referenced within the area of interest. Locally, tidal areas are not fully covered by the navigation charts depth soundings. To overcome this, the dataset





was complemented with datapoints extracted along a 500 m mesh from the National Intertidal Digital Elevation Model (NIDEM) produced by Bishop-Taylor et al. (2019).

## 5.3 Data Processing

### 5.3.1 Pre-processing

The pre-processing aims at attenuating the noise, masking the land and making sure that all meaningful information are used. It is performed in three steps, applied to the three bands (B02, B03 and B08) of all images used in subsequent processing. First, image types are converted from integer to float to make sure that cells can record decimals values. A low pass filter, corresponding to a moving average of 3 x 3 cells is then applied to minimise the effect of speckles. Lastly, the Normalized Difference Water Index (NDWI) is then calculated using the equation from Mcfeeters (1996)(Eq. (2)). Output values ranges from -1 to 1; positive values indicate the presence of open water while negative values correspond to the land. The NDWI is used to clip out areas corresponding to the land by changing negative values to null.

### 5.3.2 Derivation of the initial bathymetry

A band ratio is calculated using the first part of Stumpf et al. (2003) as presented in equation Eq. (3). The band ratio values must be tied with true water-depth measurements to determine the gain m1 and the offset m0 of equation Eq. (1). This is the key step of the data processing, as it directly impacts the validity of the generated bathymetry.

$$Band\ ratio = \frac{Ln(Blue)}{Ln(Green)} \qquad (3)$$

Where Blue corresponds to Sentinel band B02 and Green to Sentinel band B03.

To do so, band ratio values are extracted at the calibration point locations and grouped by identified depth, rounded to the first decimal. For each unique depth, the band ratio values are filtered using the interquartile range score to remove outliers, and averaged. The resulting averaged values are then plotted against the water-depth measurement from the calibration points. This reveals a linear correlation between the band ratio values and the calibration depth, up to a certain depth which is referred to as the depth of extinction. The depth of extinction is different for each satellite image and varies depending on environmental factors such as the met-ocean conditions and the turbidity of the water. To allow the batch processing of satellites images, the determination of this parameter was automated via python scripts and the use of a threshold coefficient of correlation (Fig. 7): a linear regression is calculated using all data points; if its coefficient of correlation is higher than a specific threshold, the regression is validated, else it is recalculated using all water depth, minus one meter. The process is repeated until the target coefficient of correlation is achieved. Similarly, if the targeted coefficient cannot be reached, the threshold is iteratively lowered. Ultimately each satellite image is associated to a depth of extinction and a coefficient of correlation.

The gain and the offset of the validated linear regression, which correspond to the parameters m0 and m1 of the Stumpf et al. (2003) equation, are applied to the band ratio to derive the bathymetry. These steps were performed for each satellite image.



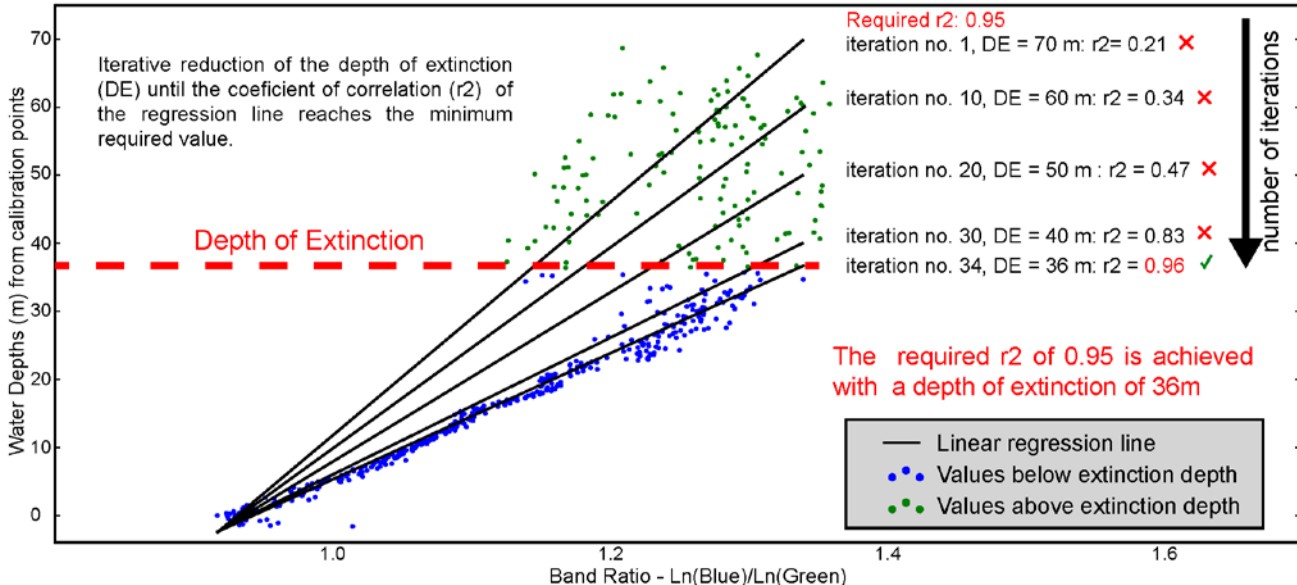

**Figure 7: Illustration of the iterative process used to determine the depth of extinction of each satellite image and the resulting regression line. The gain and the offset of this line correspond to the parameters m0 and m1.**

### 5.3.3    Correction of the initial bathymetry

The generation of the bathymetry is based on values of gain and offset averaged at the scale of satellite image tiles (100 x 100 km). Thus, the output fails to capture semi-regional (kilometric to multi-kilometric) changes of the seabed reflectance properties (Fig. 8, a). Such variation of the seabed reflectance follows trends; hence it is possible to correct its effect via the generation of an error model.

Predicted depth values from the initial bathymetry are extracted at the calibration point locations in order to calculate the absolute error between the predicted depths and the actual water depths. A regional grid of the absolute error is then generated using inverse distance weighted interpolation algorithm, where the weights are proportional to the inverse distance raised to the power p (Fig. 8, b). The intent being to capture semi-regional trends and not local variations, low values of p (0.5) were used in the interpolation, meaning that the weight of distant points is maintained. Bin sizes are calculated for each error grid grid using the root square of the average point density and are typically between 500 m and 5000 m. The resulting grid, which highlights vertical offsets related to semi-regional variation of the seabed reflectance, is then resampled to the satellite image resolution (10 x10 m) using bi-cubic convolution and added to the generated bathymetry to obtain a corrected bathymetry (Fig. 8, c).

Finally, values below the extinction depth are removed from the images. In order to take into account the vertical uncertainty of the data, the mean absolute error of the regression line is added to the depth of extinction meaning that if an image has a depth of extinction of 29 m and a mean absolute error of 1 m, all values below 30 m are removed.

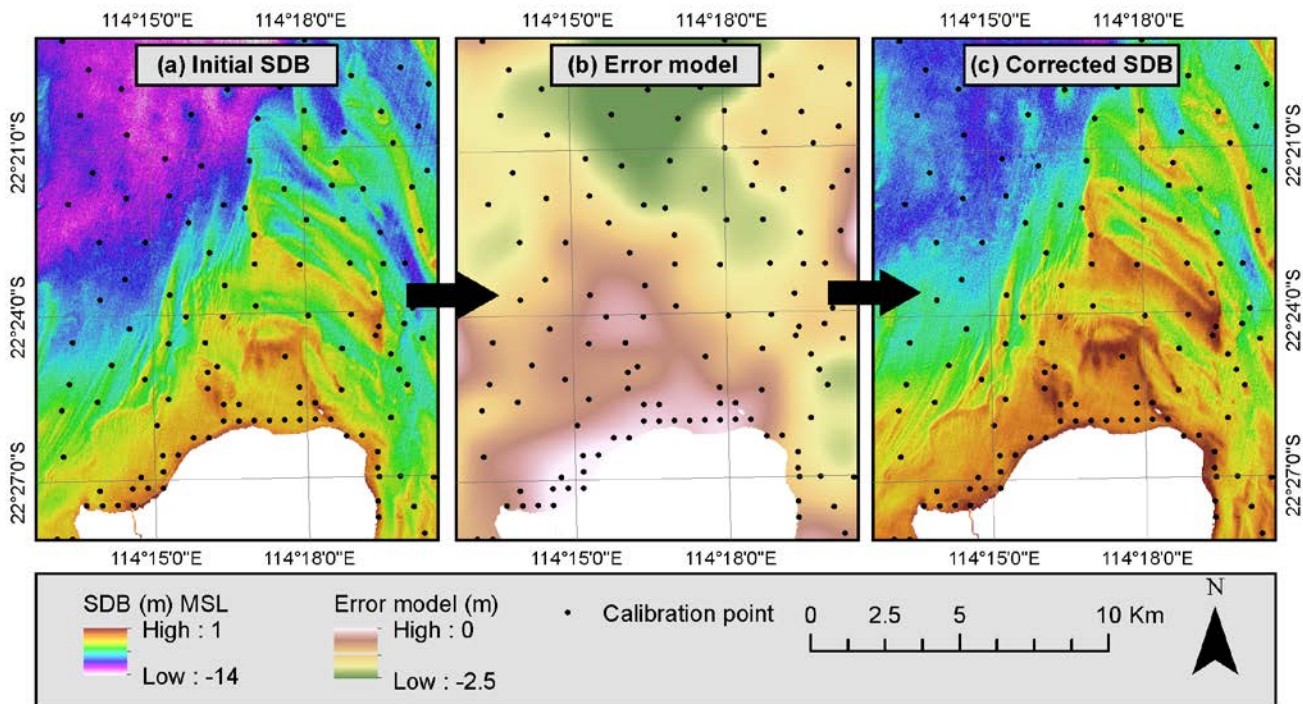

**Figure 8: illustration of the correction process applied to all seismic images. The initial SDB (a) contains errors due to semi-regional spatial variation of the seabed reflectance. An error model between the original SDB and the calibration points is calculated to illustrate these variations (b). The model is subsequently added to the original SDB to generate the corrected SDB (c).**

### 5.3.4 Generation of the bathymetry stack

The processing steps to derive the bathymetry from the satellite data described in previous sections led to the generation of 1,170 individual corrected bathymetry grids, each corresponding to the snapshot of a specific geographic area (tile) at a specific date (date of acquisition). Each grid is potentially affected by multiple temporal events and / or objects such as sediment plumes, waves or ships which result in abnormal depth values (Fig. 9, a and c). We performed statistics on the cell values from the overlapping bathymetry grids to determine the most likely depth value of a given cell, hence limiting the effect of such temporal events.

For each tile, a minimum coefficient of correlation between the predicted depth and the calibration points is determined. The threshold varies from one tile to another to reflect their respective specificity: a tile located in front of a delta, where the seabed is rapidly changing, will have overall lower coefficient of correlation values than an area with no sediment supply. On average, the threshold is set at 85%. Images with a coefficient below that threshold are discarded. In total, 222 images from 26 tiles met their respective selection criteria.

The median value of all selected grids is then calculated for each cell. The compilation of these values corresponds to the final bathymetry (Fig. 9, b and c).



**Figure 9: Illustration of the stacking process near Port Hedland (tile "KPC"). In the area, 7 images met the specified coefficient of correlation threshold and were kept for the stacking. Each image is affected by temporal effect (see example a and cross section c-c', grey and blue profiles). The median value of all images is then calculated resulting in the removal of temporal artefacts (b, cross section c-c', red profile).**

### 5.3.5   Post processing

The water land separation is based on the NDWI, meaning that any open water area was processed, including lakes and rivers. These water bodies which are often not connected to the ocean, show very variable reflectance patterns, and lack calibration points. This combination of factors often results in abnormal water-depth values. To minimize the occurrence of such artefacts, water bodies disconnected from the main ocean were removed from the final data (Fig. 10).

Band ratio values were calibrated using MSL reduced measurements, it is therefore not necessary to correct the data for tide vertical effect. However, since satellite images were acquired at different tides, sharp spatial contacts can appear along the coast at the interface between two images acquired at different tides; the spatial area covered by the ocean is different on every image (Fig. 10). These contacts were manually smoothed (only few were identified on the dataset).





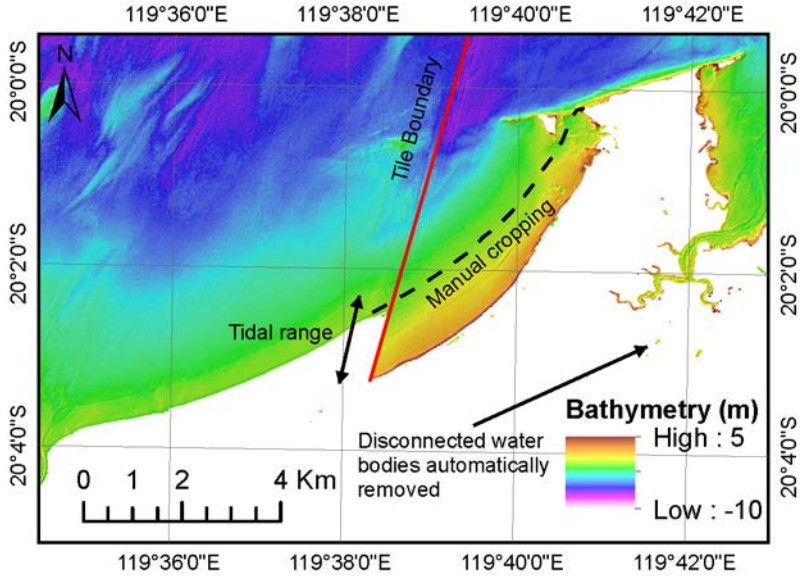

**Figure 10: illustration of the post processing filtering. Because each image is captured at a different tide, spatial mismatches can be observed near tile boundaries. Such artefact only occurs where images included in the stack do not capture the full tidal range. Upon review, they are removed manually. All disconnected water bodies were automatically removed.**

### 5.4    Data Accuracy

The satellite-derived bathymetry is compared with ~4,200 km² of Fugro LADS data (acquired from the West Australian Government's Department of Transport) in order to independently evaluate its vertical accuracy (Fig. 11, a). The comparison is based on a mesh of 500 m used to extract values from both the LADS and the SDB. Values are then plotted against each other on a density chart. Overall, there is a good correlation between the SDB and the LADS with metrics indicating a coefficient of correlation of 91% and a mean absolute error of 1.26 m (Fig. 11, b).

It is nevertheless possible to observe that some points deviate from the general trend (Fig. 11, cross-section C from 120 to 140km). Upon review of the data, it appears that these points correspond to areas where there is a constant vertical shift between the calibration points and the LADS data and therefore between the SDB and LADS. These areas were subsequently cropped out from the comparison area resulting in a virtually improved correlation between the SDB and LADS data, marked by a coefficient of correlation of 95% and a mean absolute error of 1.01 m. This highlights the importance of the calibration points

in the process and indicates that the vertical accuracy of the SDB is constrained by the vertical accuracy of the calibration points.

The LADS data coverage only represents a fraction of the 45,000 km² of processed satellite-derived bathymetry. To estimate the accuracy of the SDB elsewhere, we calculated for each pixel the standard deviation of all bathymetry grids included in the final stack. The resulting mean standard deviation is of 1.13 m, which is consistent with the accuracy calculated by comparing

the SDB and the LADS data.

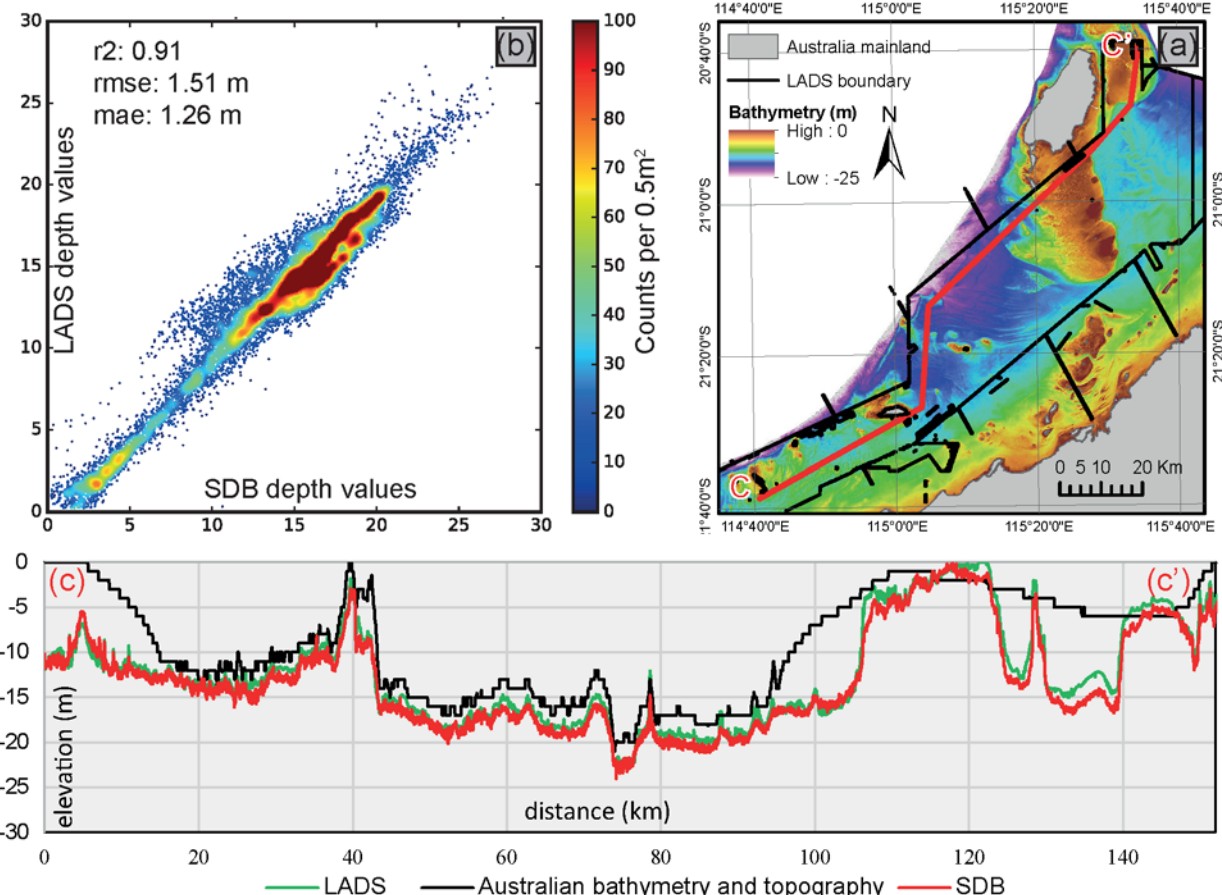

**Figure 11 Assessment of the accuracy of the SDB using LADS data. In the Vicinity of Barrow Island and Onslow, 4200 km² of LADS data overlap the SDB (a). Over 16,000 points were extracted along a mesh from both datasets and plotted against each other to assess the accuracy of the SDB (b). There is a strong correlation between both datasets as highlighted by the cross-section c. Some points**
**seem to diverge from the main trend and reflect areas with a constant shift between the SDB and LADS bathymetry (cross section c-c', km 125 to 140).**

The worst-case scenario of an average vertical error of 1.26 m indicates that the vertical accuracy is comprised between 1 m +2% of the depth and 1 m + 5% of the depth, depending on the water depth. This, combined with the positioning accuracy of the SDB of 20 m (positioning accuracy of the satellites images (Esa, 2020)), indicates that the datasets could potentially meet
the criteria of one of the assessed Zone of Confidence as defined by the S-57 Standard (Iho, 2014).

## 5.5    Data Limitation

The relative water-depth changes are linked to the resolution and to the quality of the satellites images but the absolute water-depth values are related to the calibration points, meaning that the absolute vertical accuracy of the SDB is tied by the vertical accuracy of the calibration points. Furthermore, the generation of a meaningful error model requires a minimum number of



calibration points depending on the regional water-depth changes: if the vertical offset between two adjacent calibration points is too high, the error model may generate more artefacts than it removes. This was observed, for example, west of Exmouth. The error model was consequently reviewed for each tile and removed when deemed inaccurate. Overall, the error model was kept for 25 out of the 26 satellite tiles included in the area of interest.

Finally, the stacking process aims at removing the noise and temporal artefacts by combining multiples images that were

acquired at different date. This process is overall very efficient, but if the seabed has changed drastically over the sensing period, the mobile bed-forms will be perceived by the algorithm as artefacts and will be smoothed out. Such behaviour has been observed at the mouth of the De-Grey delta, east of Port Hedland, where distributary mouth bars seem relatively active. Nevertheless, even at this location, the advantages of the stacking process surpass its limitations as it filtered out most of the water turbidity.

The computing power and the storage capacity available to the research project limited the number of satellite images that could be efficiently processed. In that regard, images were carefully selected using climate data and cloud coverage thresholds to ensure the best images were considered. Nevertheless, including the complete Sentinel catalogue and using cloud masks, hence allowing the inclusion of partly cloudy images, would presumably enhance the robustness of the model. Similarly, the current workflow solely builds on one band ratio. Recent work from Caballero and Stumpf (2019) suggests that the inclusion

of multiple band ratio can improve further the output.

## 6   Merging strategy

The creation of the regional digital elevation model is based on the integration of the newly produced satellite and seismic-derived bathymetry with the previously existing datasets. All datasets were reduced to MSL and re-sampled on a 30 m grid using bilinear interpolation. Selected datasets were included and ordered as specified in the following list (Fig. 12, Fig. 13 a):

1. Satellite-Derived Bathymetry
2. MBES Bathymetry, reprocessed from XYZ provided by GA
3. MBES Bathymetry, downloaded from AusSeabed website
4. Seismic-Derived Bathymetry
5. SRTM Digital Elevation Model
6. 2009 Australian bathymetry and Topography grid

The NIDEM, Fugro LADS data and the depth soundings from ENC tiles were not included in the final merge as they were actively used in the calibration and validation process of the other datasets. The vertical and position accuracy of the dataset used to generate the regional digital elevation model are specified in Table 3. Accuracy values were obtained using the sensing

tool specifications and datasets metadata (Iho, 2014; Esa, 2020; Rodriguez et al., 2005; Whiteway, 2009) or were conservatively estimated using intersecting high-resolution datasets (i.e. LADS, MBES).



**Table 2 Accuracy of the datasets included in the regional DEM**

| DEM | Satellite derived | Seismic derived | MBES | SRTM | 2009 Bathymetry |
|---|---|---|---|---|---|
| Source Bin size (m) | 10 | 30 | 1-40 | 30 | 250 |
| Vertical Accuracy (m) | 1 + 5%d | 1 + 2%d | 1 + 2%d | 16 | Variable |
| Position Accuracy (m) | 20 | 200 | < 30 | 20 | 250+ |

**Figure 12: Lineage of the datasets included in the regional bathymetry compilation**



## 7    Summary and outlook

The research project led to the creation of a regional 30 m digital elevation model over the Rowley Shelf and the adjacent plateaus. The dataset is based on the compilation of publicly available elevation measurements with 3D seismic and satellite-derived bathymetry produced using an innovative workflow and correspond to a major upgrade of the pre-existing regional

Australia Bathymetry and Topography grid (Fig. 13, b and b'). The vertical and positioning accuracy of the underlying datasets have been extensively assessed using high-resolution MBES. A technical committee from Geoscience Australia reviewed the data and approved it for release.

The produced dataset reveals submerged morphologies at a scale and a resolution never achieved before on the North West Shelf, allowing the onset of a wide range of local and regional studies. Marine habitat mapping and oceanographic research

rely heavily on high-resolution bathymetry. So far, such studies were limited to small areas, extrapolated afterwards to the rest of the shelf (Lyne et al., 2017; Brooke et al., 2017).  Similarly, several attempts have been made to reconstruct the paleo-geographic evolution of the shelf using low-resolution bathymetry (Larcombe et al., 2018; Ward et al., 2013; Whitley, 2017). In both cases, the inclusion of a regional high-resolution bathymetry will support the elaboration of large-scale data integration and interpretation. In addition, recent studies have investigated the presence of submerged archaeological sites along the shelf

(Benjamin et al., 2020; O'leary et al., 2020; Dortch et al., 2019). The understanding of past coastal environments stemming from the study of this dataset could steer the identification of such sites of interest (e.g., O'Leary et al., 2020).

The digital elevation model provides the full picture of modern sedimentary systems, from the source of the sediments (e.g., fluvial and carbonate systems) to their accumulation grounds along the shelf and in the basin. Parts of these modern systems can be used to better understand past sedimentary record (Paumard et al., 2020; Nyberg and Howell, 2016; Ainsworth et al.,

2019), the geotechnical properties of marine sediments (Beemer et al., 2019; Beemer et al., 2018; Senders et al., 2013) or the geohazards affecting the area (Lane and Tyler, 2015; Hogan et al., 2017; Scarselli et al., 2019; Hengesh et al., 2013). Therefore, the integration of regional high-resolution bathymetry can constitute a step change and allow researchers to ponder their results with regard to the regional context and hence support the development of more reliable interpretations and models.

The workflows presented in this paper to generate the bathymetry compilation are exclusively building on publicly available

data meaning that the method can be readily applied elsewhere in Australia and around the world.  Additional datasets will be included in the compilation as they become publicly available.



**Figure 13 Inset a displays the bathymetry compilation produced from the integration of the seismic-derived bathymetry, satellite-derived bathymetry, MBES bathymetry (from GA and reprocessed) and SRTM topography. Gaps are filled with the 9 arc seconds Australian Bathymetry and Topography grid. The compilation is a major upgrade of the pre-existing regional data (inset b vs inset b').**

## 8    Data availability

All datasets can be downloaded at: https://doi.org/10.26186/144600 (Lebrec et al., 2021). The repository includes the bathymetry compilation and the seismic-derived bathymetry with a resolution of 30 x 30 m as well as the satellite-derived bathymetry with a resolution of 10 x10 m. The repository also contains grids presenting the standard deviation and the image count of the satellite-derived bathymetry. All grids are supported by a lineage file and metadata files. Other datasets presented in this paper can be accessed through their respective references and Geoscience Australia AusSeabed data portal.

## 9    Author contribution

UL was responsible for the conceptualization of ideas and the formulation of research objectives. UL developed the methodology, collected, and processed the satellite and navigation data, and produced the bathymetry grids. VP collected the 3D seismic surveys and extracted their first reflection. MJO and SL provided guidance to organise the research. UL, VP, MJO and SL were responsible for writing, reviewing, and editing the manuscript.

## 10    Competing interests

The authors declare they have no conflict of interests.

## 11    Acknowledgements

The authors would like to thank Robert Parums from Geoscience Australia for his guidance on the collection and processing of navigation files and Anne Worden from the Australian Hydrographic Office for providing the maritime charts. The authors are also grateful to the AusSeabed Community including Kim Picard, Michele Spinnocia, Cameron Mitchel and Stephen Sagar from Geoscience Australia and Robin Beaman from James Cook University for taking the time to perform an independent review of the datasets and to Maggie Arnold for uploading and managing the grids on AusSeabed data portal.

We are grateful to Geoscience Australia and the Western Australian Department of Mines, Industry Regulation and Safety for providing the open-file 3D seismic data used in this compilation. Thanks are also due to Eliis and Esri for providing the PaleoScan™ and ArcGIS software, respectively.

Finally, the authors are thankful to the Norwegian Geotechnical Institute and the University of Western Australia for providing the funding of the project.



## 12   Financial support

The research project was conducted with the funding from the Norwegian Geotechnical Institute Ph.D. Scholarship and the University of Western Australia Scholarship for International Research Fees (SIRF).

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
