# Peer review of "Towards a regional high-resolution bathymetry of the North West Shelf of Australia based on Sentinel-2 satellite images, 3D seismic surveys and historical datasets."

_Earth System Science Data, 2021_

## Referee Comment (RC2)

**Review of paper 'Towards a regional high-resolution bathymetry of the North West Shelf of Australia based on Sentinel-2 satellite images, 3D seismic surveys and historical datasets'**

General comments:

This is a well-structured, well-illustrated manuscript that is relatively easy to follow. This paper provides the methodology for the integration of multi-source bathymetry datasets to develop a regional-scale 30 m pixel bathymetry model, spanning a depth range of coastline to over 6000 m. Such an integration of multiple bathymetry datasets is not a trivial exercise and this manuscript takes us through each of the datasets used and/or their development.

Regards the source datasets, all are useful and the methodology used to create them are well explained. Some methodologies are in widespread use, such as the Stumpf empirical method for satellite derived bathymetry data. This paper demonstrates additional innovative steps using error models to spatially fine-tune SDB results, and also using stacked images to overcome the temporal bias that occurs when just using a single satellite image. The final SDB data as the 10 m resolution nearshore bathymetry model is impressive, and will no doubt become very useful to end users.

Similarly, the seismic derived bathymetry methodology explained here is very useful as such 3D seismic data becomes more and more available. The wide use of 3D seismic first return reflectors and navigation bathymetry, as well as the offset adjustment between adjacent lines is a demonstration of best-practice, with the exception of the assumption of using 1500 m/sec for reflection derived bathymetry. I strongly recommend reprocessing reflection derived bathy using an average sound velocity for the study area.

The final regional 30 m bathymetry model will be very useful for the reasons outlined in the Summary at the end of the manuscript. But in my opinion, I think the omission of the Fugro LADS data in the merging of all source datasets is an issue. These data are high quality and even though the LADS data were used to help calibrate other datasets, I believe it is very important that these data be used in the final 30 m grid. I strongly recommend the reprocessing of the final 30m grid to include the LADS data.

The manuscript could have done with a more thorough check of language by a native-English speaking co-author to reduce the tendency to be a bit flowery at times and to reduce the number of words used to give the same meaning. A better check would have picked up obvious errors with citations or inconsistencies in the way certain source datasets were written.

The figures are well-illustrated and demonstrate well the points made in the text. However, the labels of these figures all need checking to reflect better the main text. The captions do not always read clearly and could do with more oversight and rewriting.

Detailed comments:

**Abstract**

Line 4 – Bathymetry is the measurement of depth of water in oceans, seas, or lakes. Bathymetry can be presented in numerous ways – as point clouds of multiple bathymetry (depth) points, or as raster grid products viewable in GIS, such as digital elevation model (DEM). This paper is developing integrated bathymetry model (or grid) DEM products (at 10m and 30m resolution), so confuses the use of describing a singular bathymetry grid product (their use of 'is', 'it') and the generally accepted plural use when describing bathymetry data, e.g. bathymetry data are… (i.e. bathymetry data are always plural). I recommend changing 'High-resolution bathymetry is a critical dataset for marine geoscientists. It can be used to characterize the seafloor….' to 'High-resolution bathymetry data are critical datasets for marine geoscientists. Bathymetry data can be used to characterize the seafloor and…'. Take note of use of plural when describing (any) source bathymetry data, and only use singular if referring to an individual data grid product. Make consistent throughout.

Line 13 – Suggest changing 'large-scale' to 'regional-scale'. Be careful in the use of large-scale vs small-scale – they mean different things to different people. Using 'regional-scale' consistently throughout make it clear you are covering a wide regional area.

Line 14 – Change 'to generate a regional high-resolution digital elevation model' to '…to generate regional high-resolution digital elevation models (DEMs)'

Line 15 – Change '…led to the creation of a new high-resolution bathymetry' to 'led to the creation of new high-resolution bathymetry grid'

Line 17 – Remove 'thoroughly'

Line 18 – Change 'Multi Beam Echo Sounder' to 'multibeam echo sounder' – upper case 1st letters is not in common use.

Lien 19 – Change 'depths' to 'depth'

Line 20 - Change 'depths' to 'depth'. Change 'This dataset constitutes' to 'This 30 m dataset constitutes' – to make clear you are talking about the 30m grid and not the 10 m grid

Line 21 – Change 'costal' to 'coastal'

Line 23 – Change 'All datasets used as inputs are publicly' to 'All source datasets are publicly….'. Query 'the method is' or 'the methods are'  and 'making it'? Again, not sure if we are talking plural methods (I suspect so) or a singular method? Please check.

Line 24 – Change 'The workflow as well as the resulting bathymetry' to 'The workflow and the resulting bathymetry grids…'

Line 25 – Change 'Community' to 'community'

Line 26 – Change 'The regional digital elevation model as well as the underlying datasets…' to 'The regional digital elevation models and the underlying source datasets….'

**1 Introduction**

Line 29 – Remove 'spanning'

Line 31 – Change 'extend' to 'extends'

Line 34 – Remove 'Finally

Figure 1 – Change 'Abyssal Plain to 'Argo Abyssal Plain'. This is a place name: https://www.environment.gov.au/system/files/resources/b1760d66-98f5-414f-9abf-3a9b05edc5ed/files/nw-characterisation.pdf  Change scale font size to smaller so that 125 and 250 are not so close. Suggest inset of Australia has the label 'Australia' included within the inset. Can remove the (a) and (b) labels if you are not going to refer to this in main text or in the caption. Can remove the minutes and seconds on geographic labels to reduce clutter, as these are unnecessary.

Line 39 – Change 'Location of the study area. The area of interest engulfs the Rowley Shelf (Southern half of the North West Shelf) and the adjacent plateaus' to 'Location of the North West Shelf (NWS) study area of Australia. The area of interest covers the Rowley Shelf (southern half of the NWS) and the adjacent plateaus.'

Line 46 – Change 'Multi Beam Echo Sounder' to 'multibeam echo sounder' – Upper case 1st letters are not in common usage. Change 'As of now,' to 'Currently,'

Line 47 – I believe the statement is 'less than 25% of the Australian shelf area…'. And if you have acronymed North West Shelf to NWS, then make use of this acronym.

Line 48 – Change 'number drops below 15%' to 'mapping coverage area drops below 15%'. Change 'The integration of low-resolution and indirect datasets can however allow' to 'The integration of low-resolution and multi-source datasets can, however, allow' Was not so sure about the word 'indirect' and thought 'multi-source' (i.e. satellite- and seismic-derived bathy) might be better.

Line 49 – Change 'therefore help reduce the extent of poorly charted areas' to 'therefore help improve the extent of mapped areas'. Be careful in use of term 'poorly charted' In fact, the area could be quite well charted from a mariner's navigational safety perspective, and reflected in nautical charts used by mariners, which highlight the dangers to navigation. These DEM products in this paper are not hydrographic products and will not be used for navigating from (at least not legally). So you are not improving the 'charting' of the area (this can only be done by hydrographic surveying authorities, i.e. the AHO), but you are improving seafloor mapping coverage using multi-source datasets to create integrated DEMs. The distinction is important. See: Lecours, V., Dolan, M.F.J., Micallef, A., Lucieer, V.L., 2018. A review of marine geomorphometry, the quantitative study of the seafloor Hydrology and Earth System Sciences 20, 3207-3244. doi: 10.5194/hess-20-3207-2016.

Line 51 – Change 'data, water-depth measurements from navigation charts, LiDAR surveys, satellites data and single beam surveys' to 'data, airborne LiDAR bathymetry (ALB) surveys, satellite-derived bathymetry (SDB) data and singlebeam echo sounder surveys.'  Note use of acronyms in common use, so could continue to use these acronyms going forward, e.g. MBES, ALB, SDB, if you wish.

Figure 2 – Again, use of min' sec" in geographic labels is unnecessary. Note tick marks vary between Figure 1 (outside) and Figure 2 (none) – make consistent mapping style throughout. In legend, change to National Intertidal Digital Elevation Model. See Bishop-Taylor, R., Sagar, S., Lymburner, L., Beaman, R.J., 2019. Between the tides: modelling the elevation of Australia's exposed intertidal zone at continental scale. Estuarine, Coastal and Shelf Science 223, 115-128. doi: 10.1016/j.ecss.2019.03.006. Change 'Swath' to 'MBES', as this is how you described multibeam previously. Your legend introduces two acronyms readers may not be familiar with: LADS and SRTM. You could write these out in full in the caption, e.g. 'Laser Airborne Depth Sounder = LADS….'. Could

also change National Intertidal Digital Elevation Model to use NIDEM, but then spell out in full in caption.

Line 54 – Change 'Australian Topography and Bathymetry grid' to 'Australian Topography and Bathymetry Grid (Whiteway, 2009)' This is a citation: Whiteway, T.G., 2009. Australian Bathymetry and Topography Grid, June 2009. Geoscience Australia Record 2009/21, Canberra, Australia, pp. 46. https://ecat.ga.gov.au/geonetwork/srv/eng/catalog.search#/metadata/67703

Line 57 – Change 'shelf' to 'Shelf'. But it is not really full bathymetry coverage of the NWS – only the southern or western part of the NWS (as per your description in Line 30).

Line 58 – Change 'generate such compilation' to 'generate such a compilation'

Line 62 – In the Abstract, you describe making two grids or DEMs: 10 x 10 m in nearshore areas and 30 x 30 m elsewhere. Just needs to be made consistent what you are developing, i.e. suggest whenever you are discussing the main regional DEM, to include '30 m', such as 'regional 30 m bathymetry grid'. You also use 'Digital Elevation Model' here, whereas in line 65, you use digital elevation model – this needs to be consistent throughout. And make use of acronym DEM/DEMs.

Line 68 – Change 'seamless high-resolution bathymetry' to 'seamless high-resolution bathymetry grid'. Change 'position' to 'positional'

**2 Processing tools**

Line 71 – Change 'Datasets presented in this paper were processed' to 'Source datasets presented in this paper were processed'

Line 75 – Change 'processing step' to 'processing steps'

**3 Pre-existing datasets**

Line 79 – Change 'Australian bathymetry and topography grid' to 'Australian Topography and Bathymetry Grid' as per line 54. It is a named dataset, so is upper case 1st letters when written out in full like this. This first mention of the AusBathyTopo grid should cite Whiteway (2009), as per line 54.

Line 81 – Suggest including the pixel size of the AusBathyTopo grid = 0.0025dd. Change 'The bathymetry was generated via the integration of direct water-depth soundings' to 'This bathymetry grid was developed using all available depth soundings….'

Line 84 – Change 'second' to 'arc second'

Line 87 – Change 'ground' to 'land'. Write out SRTM in full if this is first use. Citation for SRTM is: Farr, T.G., Rosen, P.A., Caro, E., Crippen, R., Duren, R., Hensley, S., Kobrick, M., Paller, M., Rodriguez, E., Roth, L., Seal, D., Shaffer, S., Shimada, J., Umland, J., Werner, M., Oskin, M., Burbank, D., Alsdorf, D., 2007. The Shuttle Radar Topographic Mission. Reviews of Geophysics 45(RG2004), 1-33. doi: 10.1029/2005RG000183. And use of SRTM data in Australia citation is found at: https://ecat.ga.gov.au/geonetwork/srv/eng/catalog.search#/metadata/72759

Line 90 – Change 'Multi Beam Echo Sounder' to 'Multibeam echo sounder'. Upper case 1st letters are not in common use.

Line 91 – same as line 90, e.g. 'Multibeam echo sounder (MBES) bathymetry'

Line 93 – Change 'The data pack' to 'This MBES dataset…'

Line 99 – Change 'National intertidal digital elevation model' to 'National Intertidal Digital Elevation Model', as this is how it is written in Bishop-Taylor, R., Sagar, S., Lymburner, L., Beaman, R.J., 2019. Between the tides: modelling the elevation of Australia's exposed intertidal zone at continental scale. Estuarine, Coastal and Shelf Science 223, 115-128. doi: 10.1016/j.ecss.2019.03.006

Line 100 – Change 'This digital elevation model was…' to 'The National Intertidal Digital Elevation Model (NIDEM) was…'

Line 103 – Change 'comprised' to 'varying'

Line 104 – Change 'Navigation charts' to 'Electronic nautical chart (ENC) tiles' – use each paragraph to write out the full name of the source dataset and any acronym. Don't assume the reader knows what an acronym means, e.g. ENC.

Line 105 – Change 'water-depth' to 'depth' – it's obvious we are talking about water depths, so no need to say this – is not in common use to use 'water-depth'

Line 106 – Change 'Point's density' to 'ENC depth point density'

Line 107 – Change 'is comprised' to 'varies'

Line 110 – Change 'AUSCoastVDT' to 'AusCoastVDT'. May want to provide a link to this: https://www.icsm.gov.au/publications/australian-coastal-vertical-datum-transformation-auscoastvdt-software

Line 111 – Change 'vertical datum' to 'vertical datums'

Line 114 – Change 'Open source LADS' to 'LADS airborne LiDAR bathymetry'

Line 115 – Change 'data was collected from' to 'data were collected by' – Note, data are plural

Line 116 – Change 'Fugro' to 'Fugro LADS Corporation'

**4 Seismic-derived bathymetry**

Line 120 – Change 'extensively surveyed, with' to 'extensively surveyed using 3D seismic techniques'

Line 122 – Change 'The bathymetry' to 'The bathymetry data'

Line 123- Change 'water-depth' to 'depth'

Line 124 – Looking at Fig 3, here you could introduce the 'reflection-derived and navigation-derived bathymetry', which is how this is written in caption for Fig 3. Make it easy for the reader to understand which is which.

Line 126 – Seems this 4.2.1 is superfluous – could just be incorporated into 4.1 as part of the overview then go straight into Data processing.

Line 127 – Remove 'Open-file' - you have already stated these are publically available in 4.1 and repeat again twice in this paragraph.

Line 130 – Change 'The extraction of the seabed reflection' to 'The reflection-derived bathymetry extraction'

Line 131 – Is 'interpretation' the right word here? Shouldn't it be 'extraction'?

Line 135 – I think this is a problem by assuming a SV value of 1500 m/sec. It is relatively easy to obtain SV profiles using CSIRO climatology data for anywhere in Australian EEZ (such as using CSIRO software SVPBuilder). Checking on a position 20deg S 115deg E in your area of interest, I get SV profiles:

0.00 1543.14
10.00 1543.22
20.00 1543.35
30.00 1543.43
50.00 1540.95
75.00 1535.93
100.00 1531.70
125.00 1527.68
150.00 1523.87
200.00 1517.29
250.00 1510.79
300.00 1505.05
400.00 1497.23
500.00 1491.57
600.00 1488.88
700.00 1488.15
800.00 1487.15
900.00 1487.04
1000.00 1487.22

Assuming you may only be interested in depths to 500 m, then this is average of 1525.371 m/sec. This can translate into underestimating the true seafloor depth by using an (incorrect) lower SV value of 1500 m/sec. For example, say you get an observed TWT of 1.0 sec. Using 1500 m/sec this translates to 750 m depth. Using a more accurate 1525 m/sec results in a depth of 762 m. The 1500 m/sec SV underestimates depth by 12.5 m, or about 1.63% in error. If this reflection-derived bathymetry is to become more common in use, such as around the Australian margin, I strongly recommend that an SV profile be generated at areas of interest which more accurately reflects the SV profile at that site. I would approach this problem by using the average SV profile value for the depth of water being studied, e.g. if depths are to 500 m, then average SV profiles (from SVPBuilder) to 500 m, and use that SV for the site. An accurate SV is fundamental to the use of singlebeam and multibeam echosounders to measure water depth, and so seismic also being sonar, should aim for similar accuracies. I do not think assuming 1500 m/sec achieves this accuracy, as in tropical waters, SV can be approaching >1540 m/sec. Assuming an SV of 1500 m/sec can underestimate true water depth and is more obvious the deeper the seafloor being measured. I think these reflection-derived bathy datasets should be recomputed with a more accurate SV profile. The Power and Clark (2019) (Line 137) work used such more accurate SV profiles and I believe you should too.

Line 150 – Very true, but the use of a constant SV is widely used in SBES surveys and is also reasonable for use in seismic-derived bathymetry. Using a more accurate SV profile for the water body area being studied will result in a more accurate first reflector derived depth, albeit the vertical resolution limits (noise) described here in water depths less than ~150 m. So having described this noise, here is a good place to explain how you dealt with the noise, i.e. rejected any seismic–derived bathy shallower than 150 m. How did you deal with this noise?

Figure 4 – Y axis is elevation – should be depth

Line 153 – Captions need to refer to (a) (b) (c). Is better to label your figures (a) (b) etc. but use 1-1', 2-2' for profiles. Currently, you use both letters for both – confusing for reader. Change 'Reflection data is' to 'Reflection data are'. Not sure what the 'until km 10' means – all looks smoothed across whole profile, but this is to be expected, as per Power and Clark (2019).

Line 158 – Change 'P1/90 files are generated' to 'P1/90 files were generated – generally use past tense in this regard.

Line 159 – Change 'water-depth' to 'depth'

Line 163 – Change 'water-depth' to 'depth'

Line 165 – provide the NOPIMS data portal link, either here in the main text or as a web reference in the References.

Line 169 - Change 'water-depth' to 'depth'

Line 171 - Change 'water-depth' to 'depth'

Line 174 – Change 'sounder location and if not, to' to 'sounder location, and if not, to'

Line 176- Change 'water-depth' to 'depth'. Change 'navigation files are then' to 'navigation files were then'

Line 178 – Change 'In this case, this formula' to 'However, this formula'

Line 181 – Change 'adapted' to 'modified'

Line 183 – Change 'comprised' to 'ranging'

Line 184 – Change 'This' to 'The'

Line 187 – Change 'increase' to 'increases'

Line 188 - Change 'water-depth' to 'depth'

Line 190 – Very true, so where you have control over the seismic-derived bathy using more accurate SV values for your area of interest, then it is imperative that you use these.

Figure 5 – labels are wrong, i.e. caption reads about filtering. But labels here are With/Without pre-processing

Line 195 – The caption reads odd, e.g. '(a) or filtered', but the label used in (a) is 'Without pre-processing'. Check this and make clear to reader.

Line 200 – Change 'seismic data is' to 'seismic data are'

Line 203 – You previously used 'Mean Sea Level' and provide acronym MSL. Be consistent. Change to 'the MSL vertical datum'

Line 209 – Change 'many areas' to 'many seismic survey areas'.  And 'not intersected by neither the ENC data points nor the MBES' seems like a double negative to me

Line 215 – Change 'only the value from the most reliable survey was' to 'only the values from the most reliable surveys were'. Change 'final merge' to 'final merged grid'

Line 217 - Change 'water-depth' to 'depth'

Line 219 – 'above'? Do you mean deeper? Be clear about using relative terms: best to use deeper/shallower, then there is no confusion to the reader

Figure 6 – Make geographic labels/tick marks etc. consistent across all figures. 'Regional data' is mislabled – should be 'AusBathyTopo data'. The (d) x and y axis labels are hard to read. Spell out r2, mae etc. Don't expect readers to automatically know what this mean, or write in the caption what MAE means.

Line 225 – Change 'accuracy of the latest' to 'accuracy of the dataset'. Change 'produced bathymetry marks' to 'extracted bathymetry data marks'

Line 226 – Include citation: Whiteway (2009)

Line 229 – Remove 'publicly available'. Use NWS

Line 230 – Do you mean 1000 m x 1000 m mesh?

Line 231 – You could quote the R2 value of 1.0 in your results (in Fig 6D)

Line 232 – Change 'Mean Absolute Error' to 'mean absolute error'. Change 'with the water depth' to ' with increasing depth,'

Table 1 – Remove 'Water'.

**5 Satellite-derived bathymetry**

Line 240 – Change 'physical' to 'physics'

Line 246 – Change 'GEBCO cook book (Iho, 2018)' to 'GEBCO Cook Book (IHO-IOC, 2018)', but is actually 2019. Can provide a link to: https://www.gebco.net/data_and_products/gebco_cook_book/

Line 247 – Remove 'in great lengths'

Line 252 – This 'Where m1…' does not read well. 'Blue and Green' should at least be 'Blue and Green bands'. Rewrite.

Line 257 - Change 'water-depth' to 'depth'

Line 262 – Remove 'Following the guidelines from the IHO (Iho, 2018),'. Change 'the GEBCO workflow' to 'The GEBCO Cook Book workflow'

Line 264- Change 'satellites' to 'satellite'

Line 268 – Change 'GEBCO like' to 'GEBCO-like'

Line 273 – Change 'directions' to 'direction'

Line 279 – Change 'aim at:' to 'aim to:', then remove the ing from the following list, e.g. 'Improving' to 'Improve' and so on

Line 287 – Do you mean Sentinal-2?

Line 291 – Not sure if Earth or earth? - check

Line 292 – Change '(Esa, 2020)' to '(ESA, 2020)'

Line 293 – Change 'highest resolution' to 'highest-resolution'

Line 294 – Change 'Sentinel' to 'Sentinel-2'

Line 297 – Change 'Ioccg' to 'IOCCG'

Line 302 – Provide the link to level-2a products

Line 307 – Change 'Bom' to 'BOM'

Line 313 - Change 'water-depth' to 'depth'

Line 315 - Change 'water-depth' to 'depth'. Use acronym NWS

Line 321 – Change 'at attenuating' to 'to attenuate'

Line 323 – Change 'decimals values' to 'decimal values'

Line 324 – Change 'speckles' to 'speckle'

Line 325 – Change '(1996)(Eq. (2)' to '(1996; Eq. 2)' – this is the better way of writing two sets of brackets.

Line 326 – Change 'The NDWI is' to 'The NDWI was'

Line 329 – Change 'A band ratio is' to 'A band ratio was'

Line 330 - Change 'water-depth' to 'depth'

Line 333 – Use 'Sentinal-2' and be consistent throughout

Line 334 – Change 'To do so, band ratio values are' to 'Band ratio values were'. Change 'depth, rounded' to 'depth and rounded'

Line 335 – Change 'values are' to 'values were'

Line 336 - Change 'water-depth' to 'depth'

Line 343 – Change 'water-depth' to 'depths'

Line 344 – Change 'coefficient of correlation' to 'coefficient of correlation r2 of >0.95'

Line 346 – Change 'are applied' to 'were applied'

Figure 7 – Change Y axis label to 'Depth (m) from calibration points'

Line 351 – This sentence does not read well – gain and offset pertain to the original satellite imagery collection, and not to do with generation of bathymetry. Rewrite.

Line 352 – I have a problem with this conclusion 'Thus, the output fails' – The (empirical) Stumpf method in general does not take into account seabed reflectance or the variability in seafloor habitat cover, e.g. the band ratio method can generate similar ratios for deep areas and those sites where the seafloor habitat cover is dark in colour. It is not to do with gain and offsets from original satellite image collection itself but rather the (relatively simple) Stumpf calculation. Frankly, I think 'Such variation' in line 353 is due to many different variables: water column turbidity, seafloor reflectance variation etc. This sentence needs rewriting. That said, the use of an error model to spatially fine-tune the original derived SDB output from a satellite image is valid and an innovative correction process. Any additional step that can be used to 'fine-tune' the final data to be more accurate is worth pursuing.

Line 355 – Change 'Predicted depth values from the initial bathymetry are' to 'Predicted depth values from the initial SDB data were'

Line 356 - Change 'water-depth' to 'depth'. Change 'error is' to 'error was'

Line 359 – Change 'Bin sizes are' to 'Bin sizes were'

Line 361 – I disagree it is just variation just due to 'seabed reflectance' as explained earlier. Just use 'error' instead of 'seabed reflectance'. Change 'is then' to 'was then'

Line 364 – Change 'are removed' to 'were removed'

Line 365 – Change 'line is' to 'line was'

Line 365 – Change 'below 30 m are removed' to 'deeper than 30 m were removed'

Line 368 – Change 'all seismic images' to 'all Sentinal-2 images'. I think this caption needs rewriting, as the errors across a SDB dataset are not just due to seafloor reflectance factor alone (see above). Change 'The model is subsequently added to the original SDB to generate the corrected SDB (c).' to 'The model is subsequently added to the original SDB (a) to generate the corrected SDB (c).

Line 372 – Change 'The processing steps to derive the bathymetry from the satellite data' to 'The processing steps to derive the corrected bathymetry data from the satellite images'

Line 374 – Change 'date' to 'date/time'

Line 375 – This is the first use of 'We' – best not to start now. Can be written 'Statistics were performed'

Line 378 – Change 'is determined' to 'was determined'

Line 380 – Change 'threshold are' to 'threshold were'

Line 383 – Change 'grids is then' to 'grids was then'

Line 384 – Change 'bathymetry' to 'SDB data after stacking'

Figure 9 – Label in (c) should be 'Bathymetry profiles'. Avoid using profiles with letters e.g. C-C' when numbers are better, e.g. 1-1', otherwise confusing against figures using same lettering, e.g. (a) (b) etc.

Line 388 – Change 'images is then' to 'images was then', and take note of my comment about labelling cross sections above. In general, all these figure captions need to be revisited to be clearer to the reader.

Line 390 – Not sure if 'Post processing' is best heading here. It is all post processing. How about 'Manual cropping'?

Line 391 – Change 'water land' to 'water-land'.

Line 393 - Change 'water-depth' to 'depth'

Line 394 – Figure 10 caption states these water bodies were automatically removed, as opposed to manually removed by cropping, e.g. the tidal zone mismatches – how were these small water bodies removed – by what process?

Line 395 – Change 'Band ratio values were calibrated using MSL reduced measurements' to 'As band ratio values were calibrated using MSL reduced depth points'  - note I cannot recall if you stated the calibration points were adjusted to MSL – just check to make sure this is stated earlier in the paper.

Line 397 – Change 'different tides' to 'different tide ranges'

Line 400 – This is not so much filtering but manual cropping. Caption needs to more accurately reflect this final step in the process.

Line 404 – Change 'The satellite-derived bathymetry is compared' to 'The final SDB data were compared'

Line 405 – Change 'evaluate its' to 'evaluate the'

Line 406 – What do you mean 'mesh of 500 m'? Change 'LADS and the SDB' to 'LADS and the SDB data'. Change 'Values are then plotted' to 'Values were plotted'

Line 407 – Change 'the SDB and the LADS' to 'the SDB data and the LADS data,'

Line 410 – Change '140km' to '140 km'

Line 411 – Change 'the SDB and LADS' to 'the SDB data and LADS data'

Line 412 – Change 'resulting in a virtually improved' to 'resulting in an improved'

Line 414 – Change 'SDB' to 'SDB data'

Line 417– Change 'SDB' to 'SDB data'. Remove the 'we' – rewrite.

Line 419 – Change 'is of 1.13 m' to 'is 1.13 m'

Figure 11 – X and Y axis labels need units (m) included. 'mae' should be 'MAE'

Line 420 – Caption needs close checking, e.g. change 'Vicinity' to 'vicinity'. Always use 'SDB data', not just 'SDB'.

Line 426 – Change 'vertical accuracy is comprised' to 'vertical accuracy lies'

Line 428 – Change '(Esa, 2020)' to '(ESA, 2020'

Line 429 – I think you are looking at the wrong IHO publication. You want S-44 Edition 6.0 0: https://iho.int/en/standards-and-specifications Your error values would likely conform to Order 2 of Table 1, and this is worth quoting in the text.

Line 430 – Change 'Data Limitation' to 'Data limitation'

Line 431 - Change 'water-depth' to 'depth'

Line 432 – Change 'SDB' to 'SDB data'

Line 434 - Change 'water-depth' to 'depth'

Line 439 – Change 'different date' to 'different dates'

Line 441 – Change 'De-Grey delta' to 'De Grey River delta'

Line 446 – Change 'Sentinal' to 'Sentinal-2'

Line 449 – Change 'improve further the output' to 'improve the results'

**6 Merging strategy**

Line 451 – Change 'model is' to 'model was'

Line 452 – Change 'on a 30 m grid' to 'to a 30 m grid, using a **** horizontal datum'. ***I see from the downloaded grid data that you use UTM50S, so state that you are using a UTM50S WGS84 projection here.

Line 459 – Change '2009 Australian bathymetry and Topography grid' to 'Australian Bathymetry and Topography Grid'

Line 461 – I think it is a mistake to exclude the LADS data, as these survey data are some of the most accurate bathymetry data available, even if for a relatively smaller area covered near Onslow. I would prefer these LADS data are included as a source dataset for the remerging of the final 30m grid. The LADS data should have a high priority in the order of source data.

Line 463 – 'sensing tool' - What is this?

Line 464 – Change '(Iho, 2014; Esa, 2020' to '(IHO, 2014; ESA, 2020'

Table 2 – Change 'Bin size' to 'pixel size'. Change '2009 Bathymetry' to 'AusBathyTopo grid'. Change 'Satellite derived' to 'Satellite-derived' etc.

Figure 12 –Make geographic labelling and style consistent across all figures. Change legend 'Bathymetry Domains' to 'Source bathymetry'.

Line 469 – Not sure that 'Lineage' is the correct word. Lineage is about origins of data. This is a figure of source datasets – not the same thing. All the captions need to be revisited.

**7 Summary and outlook**

Line 470 – Not sure if 'outlook' is best word here in the heading. Why not just 'Summary' or 'Conclusion' to draw a clear paragraph on the final results.

Line 471 – Change 'The research project' to 'This research project'

Lien 474 – Change 'Australia Bathymetry and Topography grid' to 'Australia Bathymetry and Topography Grid (Whiteway, 2009)'

Line 475 – Remove 'A technical committee from Geoscience Australia reviewed the data and approved it for release' – this still needs to get through formal review.

Line 478 – Change 'allowing the onset of a wide' to 'allowing for a wide'

Line 479 – Change 'high-resolution bathymetry' to 'high-resolution bathymetry data'. Change 'were limited' to 'have been limited'

Line 483 – Change 'bathymetry' to 'bathymetry dataset'

Line 490 – Change 'integration of regional high-resolution bathymetry grid' to 'integration of multi-source bathymetry data into a regional high-resolution bathymetry grid'. This whole sentence is a bit flowery, i.e. 'ponder the results' – rewrite.

Line 496 – Change 'data meaning' to 'data, meaning'

Figure 13 – Labels need changing, e.g. '(a) New data; overview' could be '(a) New regional bathymetry grid'; '(b) Historic data' could be '(b) Australian Bathymetry and Topography Grid

(Whiteway 2009) assuming this is the AusBathyTopo grid and so on. Actually, the colour scheme used in (a0 does not do this justice – too much of the shelf is just brown where most of the corrective effort takes place. Suggest a colour scheme that compresses more colours into the shelf region to highlight the variation in morphology on the shelf. The red box for the SDB data close-up view is too thick. And 'B' label should be 'b' to match that of the (b) Historic data (which I think is AusBathyTopo). And (b') should be (c) – this figure needs some work.

Line 498 – The caption reads incorrectly. You lead with 'Inset a displays the bathymetry' but in fact, the most important thing is the whole of the new dataset in (a), which should go in the caption first, and after then you need to mention the insets in (b). So the caption needs to be rewritten to be more precise to reference the (a) (b) and (c) figures.

**8 Data availability**

I note the data will become available on the AusSeabed Marine Data Portal if/when the paper is formally approved. Using the dropbox link, I had no issues downloaded the complete package of grids. However, I was not able to download the lineage files (8 files comprising a shapefile). There was an error and I could not open this in ArcCatalog. Recommend these separate shapefile be put inside a fodler and the data provide as a single zip file. I also don't know why these are called 'Bathymetry_Domain'. This is not in common use. Do you mean 'Bathymetry_source_data'? The Metadata document reads OK.

Robin J. Beaman

James Cook University

19 July 2021

---

## Author Comment (AC1)

Dear Anonymous Referee,

Thank you so much for taking the time to review the manuscript and to provide comments that will clearly help improving the manuscript. Please find below our response to your comments and how we plan to integrate them in the manuscript.

Best regards,

Ulysse Lebrec, on behalf of the authors.

**Digital Elevation Model (DEM) vs Bathymetry**

Throughout the manuscript there seems to be alternating use of the concept of a digital elevation model/elevation and a bathymetry model/depth. Initial description in line 63 describes a DEM, but most figures and discussion thereafter refer to bathymetry/depth. This leads to thing being mixed up (e.g. Figures 4, 9 and 11) where images use depth/bathymetry and the profile charts use elevation. Sticking with just bathymetry/depth I think would help with consistency and interpretation. I would also specify the datum in each figure (MSL).

We agree that the alternating use of bathymetry/ depth and DEM can be confusing. We will use 'bathymetry/ depth' consistently in the next revision of the manuscript apart from the compilation that will remain in 'elevation' given that it includes onshore SRTM data.

**The concept of Extinction depth**

I think the work would benefit from a better discussion around the use of the concept of Extinction depth. Firstly, a description of what this concept physically means, and how this relates to similar concepts such as Optical Depth used by other satellite derived bathymetry methods would be helpful to the non-remote sensing reader. References around these concepts and statements (e.g line 340) should be included.

Optical depth and depth of extinction refer to the same concept: the maximum depth at which the SDB is valid. It appears that authors generating SDB using the physical approach are often using the term optical depth whereas authors using the empirical method are mostly using the term depth of extinction. Physically, this means that the change in satellite images reflectance cannot be related to the water depth beyond a certain depth. We suggest doing to following modification to the text:

The resulting averaged values were then plotted against the depth measurements from the calibration points. This reveals a linear correlation between the band ratio values and the calibration depth, up to a certain depth which is referred to as the depth of extinction. The depth of extinction (sensu IHO, 2018) corresponds to the depth beyond which changes in the satellite image reflectance can no longer reflect changes in water depths, and effectively indicates the maximum depth of validity of the method. The depth of extinction varies depending on environmental factors such as the met-ocean conditions and the turbidity of the water

**Second, a bit more clarity around the target coefficient of correlation (line 344), how this is decided, and if it is the same for each image (why/why not) is needed.**

The script tries to find the depth of extinction with a minimum r2 of 0.95, if it does not work, it tries again with 0.90 etc. Suggested modification:

To allow the batch processing of satellites images, the determination of the depth of extinction was automated via python scripts and the use of a threshold coefficient of correlation (Fig. 7): a linear regression was calculated using all data points; if its coefficient of correlation r2 was higher than-a specific threshold 0.95, the regression was validated, else it was recalculated using all water depth, minus one meter. This maximum depth boundary corresponds to the theorical depth of extinction being tested (Fig. 7). The process was repeated until the target coefficient of correlation was achieved or a minimum depth of extinction of 15 m is reached. Similarly, if the targeted coefficient cannot be reached, the threshold is iteratively lowered. In such instance, the target coefficient of correlation was iteratively lowered by 0.05 and the process presented in Figure 7 (i.e., the iterative lowering of the theorical depth of extinction being tested) was repeated all over again and so forth until a target coefficient of correlation was validated. Ultimately each satellite image is associated to a depth of extinction and a coefficient of correlation.

**Filtering images in the stack and deriving the correlation coefficient**

In section 5.3.4 a process is described that essentially filters images that have outlier 'temporal effects' present (as illustrated in Figure 9). There does need to be more clarity in lines 379-383 around how a correlation coefficent threshold for each image is determined. Is there a lower threshold for an image near a river mouth with a regular sediment plume. If so, doesn't that still make that data less reliable?

The threshold was defined based on the authors judgement to find the best balance between the number of images included in the stack and their accuracy. For example, south west of Dampier all images have r2 values in excess of 0.9 whereas offshore DeGrey the highest r2 is of 0.8. This indeed suggests that the SDB generated in front of a river mouth is less reliable. The effect of a regular sediment plume is however minimised by the error model used to correct the SDB. See next comment for suggested modifications.

I think the authors also need to discuss/acknowledge how this process relates to the error correction process described in 5.3.3. As the error correction already corrects the bathymetry based on a surface error model in comparison to the calibration points, if you are then looking at a correlation model based on this corrected bathymetry, the process is at risk of becoming circular and less valid. For example, it seems that in a turbid estuary, the error model process would do its best to correct the underestimated bathy values (in a regional surface sense) back to the calibration points. Running a correlation then for image QA/QC on these already corrected outputs needs a bit more justification I think.

The coefficient of correlations used here are the ones obtained from the derivation of the initial bathymetry presented in section 5.3.2 and which are therefore calculated before the error model correction. We are not calculating any coefficient of correlation on corrected images as this would provide meaningless values and basically be, as you describe, a circular self-correlation.

We suggest doing the following modification to the text:

For each tile, a minimum coefficient of correlation between the predicted depth and the calibration points is determined and images with a coefficient below that threshold are discarded. Coefficients of correlation values used here to determine if an image should or should not be included in the stack

are the values calculated during the derivation of the initial bathymetry, before the application of any types of correction, to avoid circular correlations. The threshold varies from one tile to another to reflect their respective specificity: a tile located in front of a delta, where the seabed is rapidly changing, will have overall lower coefficient of correlation values than an area with no sediment supply. In that regard it was not possible to establish a firm rule and the threshold was subsequently determined for each tile by the authors to obtain the best ratio between the number of images integrated in the stack and their respective coefficient of correlation. On average, the threshold is set at 85%. In total, 222 images from 26 tiles met their respective selection criteria.

**Use of pixel based Standard Deviation layer**

The inclusion of a pixel based standard deviation layer in the data product is a very useful tool, and I think should be used more in the manuscript, as it is only mentioned as an afterthought in line 418 and Section 8. Already in Figure 9(c), we can see the expected increased variance in the single image solutions as depth increases. Showing an image illustrating this based on the standard deviation layer would be very informative. Likewise, an image figure would help illustrate how a higher variance of the product would be expected in dynamic and/or turbid estuaries, helping to back up statement such as line 443.

In my opinion, this SD layer is as useful in terms of the user assessing the accuracy of the SDB product as the validation to the LADS data. To add further value to the statement on mean SD in line 419, I would suggest extracting a graph/table that shows the mean SD for pixels based on depth intervals (ie. Model depth SD for pixels 0-2m, 2-5m, 5-10m etc etc). This would be extremely helpful to the end user.

Agree, interestingly one could use the STD layer to assess bedform mobility. We suggest adding (and commenting in text) the following figure and table. Overall, the standard deviation increases:

- With depth.
- In areas with potential mobile bedforms including near river mouths and tidal channels.
- In areas with high tidal range (the standard deviation increases eastward along with the tidal range. The only exception is a tile which includes an insufficient number of images in the stack to generate meaningful metrics due to the high turbidity of the area.
- In turbid/ muddy areas.
- Potentially where major change of seabed type occurs (i.e., sea grass meadows).

It should be noted that the standard deviation is however very sensible to the number of images included in the stack. Areas where only a few images were included in the stack tend to show lower standard deviation values which does necessarily means that they are more accurate.

Figure 1 Illustration of the final satellite-derived bathymetry (a) and of the associated standard deviation (b), south of Barrow island. The standard deviation is calculated for each pixel using the values from all individual bathymetry grids included in the stack. The resulting grid provides an estimate of the vertical accuracy of the final bathymetry at any given points. The standard deviation increases (and hence the accuracy reduces) with increasing water depths (b) and in dynamic environments that are changing under modern oceanic conditions such as in the vicinity of tidal sand bars and channels (b).

Table 1 Mean standard deviation per depth range.

| Depth range (m)             | 0 - 5 | 5 - 10 | 10 - 15 | 15 - 20 | 20 - 25 | 25 - 30 |
|-----------------------------|-------|--------|---------|---------|---------|---------|
| Mean standard deviation (m) | 0.90  | 1.04   | 1.11    | 1.23    | 1.18    | 1.53    |

**Technical Corrections**

*Line 49* – *Please explain 'indirect' datasets, the meaning is not particularly clear.*

As per suggestion from Robin Beaman, we will change it to 'multi-source'

*Figure 6* – In description, it should be made clearer that the Australian Bathymetry and Topography refers to the Regional Data in panels b and c.

Figure label was modified to AusSeabedTopo.

*Lines 288, 323 and elsewhere* – Including the band centre wavelengths for the Sentinel 2 bands described would be helpful.

Ok.

*Lines 305 – 312 – A rewording and perhaps further explanation I think would help to explain what is meant by abnormal values and the rational for avoiding them (ie. Dry season in the North?)*

Abnormal values refer to pixel values from the input satellite images that are affected by seasonal environmental factors (e.g., water turbidity increases during wet season) and may therefore not be representative of the water depth. We will clarify this point in the text. The rational for avoiding them is explained the next paragraph.

*Line 325* – *Please elaborate on what is meant by speckles (ie pixel based glint, signal/noise artefacts)*

Pixel based glint. Such filtering is recommended by the IHO cookbook. We will update sentence accordingly.

Figure 8 – I think 'seismic' may meant to be 'satellite'

Indeed.

Line 376 – Perhaps 'statistical analysis' instead of 'statistics'

Ok.

*Figure 9* – Would benefit from inclusion of the true colour image of this example to visually show the temporal artefacts concept the author is trying to highlight.

We can add another 'row' to the figure with three insets showing three insets from true colour image illustrating each of the three points (ships, clouds, turbid water, Fig. 2).

---

## Author Comment (AC2)

**Ulysse Lebrec**
**Ph.D. Student**
**Centre for Energy Geoscience / School of Earth Sciences**

Dear Robin Beaman,

Thank you so much for providing such in depth comments on the manuscript and the associated datasets. Please find below our response to your main comments and how we plan to integrate them.

We are very grateful for the thorough review of the English. As a non-native English speaker, improving the quality and clarity of the writing is a continuous and never-ending challenge. We will integrate your re-writing suggestions in the next revision of the manuscript, and we believe that this will greatly improve the quality of the manuscript. Similarly, we have reviewed your suggested references and will include them accordingly.

Best regards,

Ulysse Lebrec, on behalf of the authors.

Our understanding is that your two main comments are as follow:

**Include the LADS LiDAR data in the compilation**

The LADS LiDAR dataset was made available to us under Creative Commons licensing CC BY-NC-SA which forbids any commercial use of the data. Any remix or transformation of the data should also carry the same licence. On the other hand, AusSeabed data portal, used to share the compilation, is based on Creative Commons licensing CC BY hence allowing commercial use.

Our understanding is that we are therefore not allowed to include the LADS LiDAR dataset in the compilation as this would breach the LADS LiDAR dataset CC BY-NC-SA licence.

**Reprocess the seismic derived bathymetry to include refined velocity models**

In the absence of site-specific sound velocity profiles, we made the decision to use a constant value to convert seismic-derived bathymetry from the time to the depth domain instead of using more refined approaches such as polynomial equations. The value of 1500 m/s was retained by averaging the sound velocity values that were sometimes specified in the navigation file headers of the seismic surveys. Importantly, this conversion was only performed for the reflection-derived bathymetry. In the case of navigation-derived bathymetry, the input data was already in depth and, most of the time, velocities were not available at all, hence forbidding back calibration. Reflection-derived bathymetry includes 26 surveys with an average depth of circa -750 m.

You mentioned in your comments the 'CSIRO software SVP builder' that can generate synthetic SVP profiles using climatology data anywhere around Australia. After enquiring about it, it appears to be an earlier version of the Doris software https://www.doris-svp.org/ which is published by IFREMER and SHOM and uses climatology data from the World Ocean Atlas 2013 (https://climatedataguide.ucar.edu/climate-data/world-ocean-atlas-2013-woa13). To the best of our knowledge this is the only tool available to generate such synthetic profiles.

The University of Western Australia,
Perth WA 6009 Australia

**E**  ulysse.lebrec@research.uwa.edu.au
       ulysse.lebrec@ngi.no

CRICOS Provider Code 00126G

We generated a few profiles within the area of interest to compare the differences between the resulting values with the constant 1500 m/s used in the manuscript. It should be noted that the tool returns more depth/velocity pairs in shallow waters (i.e., the vertical interval between two velocity points increases with depth). We therefore filtered the synthetic profile values to obtain meaningful averages. The resulting average velocities per depth interval is presented in Table 1. This comparison indicates that for most intervals the difference is below 0.5%.

The software does not provide quantifiable uncertainties for neither the climatology data used as input nor for the computation of the velocities themselves, it is therefore not possible to fully assess by what extent the synthetic profiles would actually improve the results.

Additionally, the difference in velocities should be looked at in the context of seismic surveys vertical resolution. For the most part, seismic surveys are acquired with a vertical sampling rate of 2 to 4 ms and frequencies comprised between 40-150 hertz (MBES surveys use frequencies 1000s of times higher). Moreover, as presented in the manuscript, morphologies from the reflection derived bathymetry exhibit increasing vertical distortion in shallow waters, overwriting by an order of magnitude any offset related to the sound velocity model.

In light of the above, we believe that there are not enough elements to support a reprocessing of the reflection derived bathymetry using synthetic sound velocity profiles. The uncertainties associated with sound velocity profiles are covered by the current data limitation sections and data accuracy tables.

We certainly agree that the use of site-specific sound velocity profiles should become part of the best practice to produce reflection seismic-derived bathymetry. However, we think that the actual accuracy gain from the inclusion of synthetic sound velocity profiles should be further assessed and that such task extends beyond the scope of this paper.

*Table 1 Comparison of average velocities from Doris with a constant value of 1500 m/s*

| Depth Interval (m) | 0-250 | 0-500 | 0-750 | 0-1000 | 0-1500 | 0-2000 |
|---|---|---|---|---|---|---|
| Avg Velocity (Doris) m/s | 1526.37 | 1512.74 | 1505.2 | 1500.86 | 1496.58 | 1495.1 |
| % difference with 1500 m/s | 1.72 | 0.84 | 0.34 | 0.05 | 0.23 | 0.32 |

**Additional comments.**

we have included below additional questions raised in the PDF document. Comments and text editing suggestions that are not specifically mentioned are regarded as "accepted" and will be included in the next revision of the manuscript.

**Line 57 – Change 'shelf' to 'Shelf'. But it is not really full bathymetry coverage of the NWS – only the southern or western part of the NWS (as per your description in Line 30).**
The combination of this dataset with the compilation you produced over the Northern Territory results in a full coverage of the NWS.

**Line 126. Could you incorporate 4.1.2 with 4.1.**

While we agree that in this specific case having a sub section for data source is a bit superfluous, the idea was to have the same section breakdown for each datatype.

**Line 131. How did you deal with noise < 150 m?**

Whenever possible we used navigation-derived bathymetry instead of reflection-derived bathymetry as this issue is specific to reflection-derived bathymetry. However, in some areas, navigation data was either not available or not dense enough to generate a grid. In such case we included the reflection-derived bathymetry because we consider that, while the relative elevation of a given seabed feature compared to another one might be inaccurate, the actual morphology is still valid and adds value compared to the Australian Bathymetry and Topography grid. The bottom line is that we dealt with the noise by choosing which survey to include in the compilation, but we did not apply any specific correction to filter that noise as we could not find an accurate method to precisely quantify it. This is explained in section 4.5.

**Line 394. How were small water bodies automatically removed?**

We first generated a raster domain shapefile (a shapefile delineating the boundaries of a raster) where each feature – in this case polygons – corresponds to an individual water body. We then filtered the shapefile to only keep the polygon corresponding to the main water body and used it to clip the bathymetry, effectively resulting in the removal of all disconnected water bodies. We will add a clarification in the text.

**Line 406 What do you mean by mesh of 500 m?**

A grid (fishnet) of points separated by 500 m along the X and Y axis. We will add a clarification in the text.

**Line 429 – I think you are looking at the wrong IHO publication. You want S-44 Edition 6.0 0: https://iho.int/en/standards-and-specifications Your error values would likely conform to Order 2 of Table 1, and this is worth quoting in the text.**

The document cited in the text https://iho.int/uploads/user/pubs/standards/s-57/S-57_e3.1_Supp3_Jun14_EN.pdf refers to Zone Of Confidence (see p 17). We are happy to use S-44 instead as the point remains the same.

**Line 463 Sensing tool what is that?**

We are referring to the different types of bathymetry (e.g., seismic vs satellite vs MBES). We will rephrase to say just that.

**Data availability. Error when loading shapefile and use of the term 'bathymetry domain'.**

The shapefile is pretty heavy (1.9Gb). Given that by default Arcmap/catalogue only has 2Gb of ram this can easily make the software crash.

The term domain refers to the coverage of the datasets included in the compilation. I believe you used the term lineage in your own compilation. The term 'domain' corresponds to the name of the ArcGIS geoprocessing tool used to generate the file and is, to the best of our knowledge, commonly used in GIS.

---

## Author Response (AR1)

**Ulysse Lebrec**
**Ph.D. Student**
**Centre for Energy Geoscience / School of Earth Sciences**

*The Editors*                                                          *30 August 2021*
*Earth System Science Data*

Dear Dr. Prasad Gogineni,

Please find attached the revised version of the manuscript essd-2021-128 entitled:

**Towards a regional high-resolution bathymetry of the North West Shelf of Australia based on Sentinel-2 satellite images, 3D seismic surveys and historical datasets.**

By Ulysse Lebrec, Victorien Paumard, Michael O'Leary and Simon C. Lang.

All comments and issues raised by the reviewers were addressed including updates to the text and figures. Our response to individual comments and how we addressed them is detailed below.

**All authors have approved the updated manuscript and have agreed to its submission.**

We look forward to hearing back from you.

Yours sincerely,

Ulysse Lebrec

The University of Western Australia,          **M** +61 468 596 605          **E** ulysse.lebrec@research.uwa.edu.au
Perth WA 6009 Australia                                                              ulysse.lebrec@ngi.no
                                                                                     CRICOS Provider Code 00126G

**Comments from Referee 1.**

**Digital Elevation Model (DEM) vs Bathymetry**

*Throughout the manuscript there seems to be alternating use of the concept of a digital elevation model/elevation and a bathymetry model/depth. Initial description in line 63 describes a DEM, but most figures and discussion thereafter refer to bathymetry/depth. This leads to thing being mixed up (e.g. Figures 4, 9 and 11) where images use depth/bathymetry and the profile charts use elevation. Sticking with just bathymetry/depth I think would help with consistency and interpretation. I would also specify the datum in each figure (MSL).*

We agree that the alternating use of bathymetry/ depth and DEM can be confusing. We have used 'bathymetry/ depth' consistently in the revised manuscript apart from the compilation that remains in 'elevation' given that it includes onshore SRTM data.

**The concept of Extinction depth**

*I think the work would benefit from a better discussion around the use of the concept of Extinction depth. Firstly, a description of what this concept physically means, and how this relates to similar concepts such as Optical Depth used by other satellite derived bathymetry methods would be helpful to the non-remote sensing reader. References around these concepts and statements (e.g line 340) should be included.*

Optical depth and depth of extinction refer to the same concept: the maximum depth at which the SDB is valid. It appears that authors generating SDB using the physical approach are often using the term optical depth whereas authors using the empirical method are mostly using the term depth of extinction. Physically, this means that the change in satellite images reflectance cannot be related to the water depth beyond a certain depth. We did the following modification to the text (see line 367, with track changes):

*The resulting averaged values were then plotted against the depth measurements from the calibration points. This revealed a linear correlation between the band ratio values and the calibration depths, up to a certain depth which is referred to as the depth of extinction. The depth of extinction (sensu International Hydrographic Organization and Intergovernmental Oceanographic Commission (2019)) corresponds to the depth beyond which changes in the satellite image reflectance can no longer reflect changes in depths, and effectively indicates the maximum depth of validity of the method. The depth of extinction is different for each satellite image and varies depending on environmental factors such as the met-ocean conditions and the turbidity of the water.*

*Second, a bit more clarity around the target coefficient of correlation (line 344), how this is decided, and if it is the same for each image (why/why not) is needed.*

The script tries to find the depth of extinction with a minimum r2 of 0.95, if it does not work, it tries again with 0.90 etc. We included the following clarification to the text (see line 373, with track changes):

*To allow batch processing satellites images, the determination of the depth of extinction was automated via python scripts and the use of a threshold coefficient of correlation (Fig. 7): a linear regression was calculated using all data points; if its coefficient of correlation r2 was higher than 0.95, the regression was validated, else it was recalculated using all depths, minus one meter. This maximum depth boundary corresponds to the theorical depth of extinction being tested*

*(Fig. 7). The process was repeated until the target coefficient of correlation was achieved or a minimum depth of extinction of 15 m was reached.  In such instance, the target coefficient of correlation was iteratively lowered by 0.05 and the process presented in Figure 7 (i.e., the iterative lowering of the theorical depth of extinction being tested) was repeated all over again and so forth until a target coefficient of correlation was validated. Ultimately each satellite image was associated to a depth of extinction and a coefficient of correlation.*

**Filtering images in the stack and deriving the correlation coefficient**

*In section 5.3.4 a process is described that essentially filters images that have outlier 'temporal effects' present (as illustrated in Figure 9). There does need to be more clarity in lines 379-383 around how a correlation coefficent threshold for each image is determined. Is there a lower threshold for an image near a river mouth with a regular sediment plume. If so, doesn't that still make that data less reliable?*

The threshold was defined based on the authors judgement to find the best balance between the number of images included in the stack and their accuracy. For example, south west of Dampier all images have r2 values in excess of 0.9 whereas offshore DeGrey the highest r2 is of 0.8. This indeed suggests that the SDB generated in front of a river mouth is less reliable. The effect of a regular sediment plume is however minimised by the error model used to correct the SDB. See next comment for included modifications.

*I think the authors also need to discuss/acknowledge how this process relates to the error correction process described in 5.3.3. As the error correction already corrects the bathymetry based on a surface error model in comparison to the calibration points, if you are then looking at a correlation model based on this corrected bathymetry, the process is at risk of becoming circular and less valid. For example, it seems that in a turbid estuary, the error model process would do its best to correct the underestimated bathy values (in a regional surface sense) back to the calibration points. Running a correlation then for image QA/QC on these already corrected outputs needs a bit more justification I think.*

The coefficient of correlations used here are the ones obtained from the derivation of the initial bathymetry presented in section 5.3.2 and which are therefore calculated before the error model correction. We are not calculating any coefficient of correlation on corrected images as this would provide meaningless values and basically be, as described by the reviewer, a circular self-correlation.

We performed the following modification to the manuscript (see line 421-427, with track changes):

*For each tile, a minimum coefficient of correlation between the predicted depths and the calibration points was determined and images with a coefficient below that threshold were discarded. Coefficients of correlation values used here to determine if an image should or should not be included in the stack were the values calculated during the derivation of the initial bathymetry, before the application of any types of correction, to avoid circular correlations. The threshold varied from one tile to another to reflect their respective specificities: tiles located in front of a delta, where the seabed is rapidly changing, have overall lower coefficient of correlation values than areas with no sediment supply. In that regard it was not possible to establish a firm rule and the threshold was subsequently determined for each tile by the authors to obtain the best ratio between the number of images integrated in the stack and their respective coefficient of correlation. On average, the threshold was set at 85%. In total, 222 images from 26 tiles met their respective selection criteria.*

**Use of pixel based Standard Deviation layer**

*The inclusion of a pixel based standard deviation layer in the data product is a very useful tool, and I think should be used more in the manuscript, as it is only mentioned as an afterthought in line 418 and Section 8. Already in Figure 9(c), we can see the expected increased variance in the single image solutions as depth increases. Showing an image illustrating this based on the standard deviation layer would be very informative. Likewise, an image figure would help illustrate how a higher variance of the product would be expected in dynamic and/or turbid estuaries, helping to back up statement such as line 443.*

*In my opinion, this SD layer is as useful in terms of the user assessing the accuracy of the SDB product as the validation to the LADS data. To add further value to the statement on mean SD in line 419, I would suggest extracting a graph/table that shows the mean SD for pixels based on depth intervals (ie. Model depth SD for pixels 0-2m, 2-5m, 5-10m etc etc). This would be extremely helpful to the end user.*

Agree, interestingly one could use the STD layer to assess bedform mobility. We added the following figure and table. Overall, the standard deviation increases:

- With depth.
- In areas with potential mobile bedforms including near river mouths and tidal channels.
- In areas with high tidal range (the standard deviation increases eastward along with the tidal range. The only exception is a tile which includes an insufficient number of images in the stack to generate meaningful metrics due to the high turbidity of the area.
- In turbid/ muddy areas.
- Potentially where major change of seabed type occurs (i.e., sea grass meadows).

It should be noted that the standard deviation is however very sensible to the number of images included in the stack. Areas where only a few images were included in the stack tend to show lower standard deviation values which does necessarily means that they are more accurate. The text was updated as follow (see line 471-475, with track changes):

The standard deviation appears to increase with depth (Fig. 12, Table 2) but also in areas with strong current, water turbidity, tidal range and potentially where major change of seabed type occurs (e.g., seagrass meadows), effectively highlighting areas that have changed significantly through the time interval included in the final stack. This suggests that the standard deviation layer could be used to better understand seabed conditions and could potentially, for example, help identifying mobile bedforms.

[Figure]

*Figure 1 Illustration of the standard deviation grid (b) generated with the final SDB bathymetry (a). The grid illustrates the spatial variability of the final SDB grid accuracy. The standard deviation increases (and hence the accuracy of the bathymetry decreases) with depth as well as in dynamic environment where the seabed changed through the sensing period such as tidal passes (b).*

*Table 1 Mean standard deviation per depth range.*

| Depth range (m) | 0 - 5 | 5 - 10 | 10 - 15 | 15 - 20 | 20 - 25 | 25 - 30 |
|---|---|---|---|---|---|---|
| Mean standard deviation (m) | 0.90 | 1.04 | 1.11 | 1.23 | 1.18 | 1.53 |

***Technical Corrections***

***Line 49*** *– Please explain 'indirect' datasets, the meaning is not particularly clear.*

As per suggestion from Robin Beaman, we changed it to 'multi-source'

***Figure 6*** *– In description, it should be made clearer that the Australian Bathymetry and Topography refers to the Regional Data in panels b and c.*

Figure label was modified to AusBathyTopo.

***Lines 288, 323 and elsewhere*** *– Including the band centre wavelengths for the Sentinel 2 bands described would be helpful.*

We included it in section 5.2.1 (data selection, see line 337 with track changes).

***Lines 305 – 312*** *– A rewording and perhaps further explanation I think would help to explain what is meant by abnormal values and the rational for avoiding them (ie. Dry season in the North?)*

Abnormal values refer to pixel values from the input satellite images that are affected by seasonal environmental factors (e.g., water turbidity increases during wet season) and may therefore not be representative of the water depth. We clarified this point in the text (see line 332, with track changes). The rational for avoiding them is explained the next paragraph.

***Line 325*** *– Please elaborate on what is meant by speckles (ie pixel based glint, signal/noise artefacts)*

Pixel based glint. Such filtering is recommended by the IHO cookbook. We updated the sentence accordingly (see line 353, with track changes).

***Figure 8*** *– I think 'seismic' may meant to be 'satellite'*

Indeed.

***Line 376*** *– Perhaps 'statistical analysis' instead of 'statistics'*

Ok.

***Figure 9*** *– Would benefit from inclusion of the true colour image of this example to visually show the temporal artefacts concept the author is trying to highlight.*

We added another 'row' to the figure with three insets from true colour images illustrating each of the three points (ships, clouds, turbid water, Fig. 2; Fig. 9 in revised manuscript).

[Figure]

*Figure 2. Updated Fig. 9 (cf revised manuscript)*

**Line 433** - *use of 'constrained' rather than 'tied' perhaps*

Ok.

**Comments from referee 2:**

Our understanding is that the two main comments are as follow:

*Include the LADS LiDAR data in the compilation*

The LADS LiDAR dataset was made available to us under Creative Commons licensing CC BY-NC-SA which forbids any commercial use of the data. Any remix or transformation of the data should also carry the same licence. On the other hand, AusSeabed data portal, used to share the compilation, is based on Creative Commons licensing CC BY hence allowing commercial use.

Our understanding is that we are therefore not allowed to include the LADS LiDAR dataset in the compilation as this would breach the LADS LiDAR dataset CC BY-NC-SA licence.

*Reprocess the seismic derived bathymetry to include refined velocity models*

In the absence of site-specific sound velocity profiles, we made the decision to use a constant value to convert seismic-derived bathymetry from the time to the depth domain instead of using more refined approaches such as polynomial equations. The value of 1500 m/s was retained by averaging the sound velocity values that were sometimes specified in the navigation file headers of the seismic surveys. Importantly, this conversion was only performed for the reflection-derived bathymetry. In the case of navigation-derived bathymetry, the input data was already in depth and, most of the time, velocities were not available at all, hence forbidding back calibration. Reflection-derived bathymetry includes 26 surveys with an average depth of circa -750 m.

You mentioned in your comments the 'CSIRO software SVP builder' that can generate synthetic SVP profiles using climatology data anywhere around Australia. After enquiring about it, it appears to be an earlier version of the Doris software https://www.doris-svp.org/ which is published by IFREMER and SHOM and uses climatology data from the World Ocean Atlas 2013 (https://climatedataguide.ucar.edu/climate-data/world-ocean-atlas-2013-woa13). To the best of our knowledge this is the only tool available to generate such synthetic profiles.

We generated a few profiles within the area of interest to compare the differences between the resulting values with the constant 1500 m/s used in the manuscript. It should be noted that the tool returns more depth/velocity pairs in shallow waters (i.e., the vertical interval between two velocity points increases with depth). We therefore filtered the synthetic profile values to obtain meaningful averages. The resulting average velocities per depth interval is presented in Table 1. This comparison indicates that for most intervals the difference is below 0.5%.

The software does not provide quantifiable uncertainties for neither the climatology data used as input nor for the computation of the velocities themselves, it is therefore not possible to fully assess by what extent the synthetic profiles would actually improve the results.

Additionally, the difference in velocities should be looked at in the context of seismic surveys vertical resolution. For the most part, seismic surveys are acquired with a vertical sampling rate of 2 to 4 ms and frequencies comprised between 40-150 hertz (MBES surveys use frequencies 1000s of times higher). Moreover, as presented in the manuscript, morphologies from the reflection derived bathymetry exhibit increasing vertical distortion in shallow waters, overwriting by an order of magnitude any offset related to the sound velocity model.

In light of the above, we believe that there are not enough elements to support a reprocessing of the reflection derived bathymetry using synthetic sound velocity profiles. The uncertainties associated

with sound velocity profiles are covered by the current data limitation sections and data accuracy tables.

We certainly agree that the use of site-specific sound velocity profiles should become part of the best practice to produce reflection seismic-derived bathymetry. However, we think that the actual accuracy gain from the inclusion of synthetic sound velocity profiles should be further assessed and that such task extends beyond the scope of this paper.

*Table 2 Comparison of average velocities from Doris with a constant value of 1500 m/s*

| Depth Interval (m) | 0-250 | 0-500 | 0-750 | 0-1000 | 0-1500 | 0-2000 |
|---|---|---|---|---|---|---|
| Avg Velocity (Doris) m/s | 1526.37 | 1512.74 | 1505.2 | 1500.86 | 1496.58 | 1495.1 |
| % difference with 1500 m/s | 1.72 | 0.84 | 0.34 | 0.05 | 0.23 | 0.32 |

**Additional comments.**

we have included below additional questions raised in the PDF document. Comments and text editing suggestions that are not specifically mentioned hereafter are regarded as "accepted" and were included in the revised manuscript.

**Line 57 – Change 'shelf' to 'Shelf'. But it is not really full bathymetry coverage of the NWS – only the southern or western part of the NWS (as per your description in Line 30).**
The combination of this dataset with the compilation produced over the Northern Territory by the reviewer results in a full coverage of the NWS. As stated in the text, it is the combination of both compilations that provide a full coverage of the NWS.

**Line 126. Could you incorporate 4.1.2 with 4.1.**

While we agree that in this specific case having a sub section for data source is a bit superfluous, the idea was to have the same section breakdown for each datatype.

**Line 131. How did you deal with noise < 150 m?**

Whenever possible we used navigation-derived bathymetry instead of reflection-derived bathymetry as this issue is specific to reflection-derived bathymetry. However, in some areas, navigation data was either not available or not dense enough to generate a grid. In such case we included the reflection-derived bathymetry because we consider that, while the relative elevation of a given seabed feature compared to another one might be inaccurate, the actual morphology is still valid and adds value compared to the Australian Bathymetry and Topography grid. The bottom line is that we dealt with the noise by choosing which survey to include in the compilation, but we did not apply any specific correction to filter that noise as we could not find an accurate method to precisely quantify it. This is explained in section 4.5.

**Line 394. How were small water bodies automatically removed?**

We first generated a raster domain shapefile (a shapefile delineating the boundaries of a raster) where each feature – in this case polygons – corresponds to an individual water body. We then filtered the shapefile to only keep the polygon corresponding to the main water body and used it to clip the

bathymetry, effectively resulting in the removal of all disconnected water bodies. We added clarification in the text (see line 441-445, with track changes).

**Line 406 What do you mean by mesh of 500 m?**

A grid (fishnet) of points separated by 500 m along the X and Y axis. We added a clarification in the text (see line 457, with track changes).

**Line 429 – I think you are looking at the wrong IHO publication. You want S-44 Edition 6.0 0: https://iho.int/en/standards-and-specifications Your error values would likely conform to Order 2 of Table 1, and this is worth quoting in the text.**

The document cited in the text [https://iho.int/uploads/user/pubs/standards/s-57/S-57_e3.1_Supp3_Jun14_EN.pdf](https://iho.int/uploads/user/pubs/standards/s-57/S-57_e3.1_Supp3_Jun14_EN.pdf) refers to Zone Of Confidence (see p 17). We have however followed the reviewer suggestion to use S-44 instead as the point remains the same (see line 486, with track changes).

**Line 463 Sensing tool what is that?**

We are referring to the different types of bathymetry (e.g., seismic vs satellite vs MBES). We rephrased the sentence to use 'source data metadata' instead (see line 532, with track changes).

**Data availability. Error when loading shapefile and use of the term 'bathymetry domain'.**

The shapefile is pretty heavy (1.9Gb). Given that by default Arcmap/catalogue only has 2Gb of ram this can easily make the software crash.

The term domain refers to the coverage of the datasets included in the compilation. We believe the reviewer used the term lineage in its own compilation. The term 'domain' corresponds to the name of the ArcGIS geoprocessing tool used to generate the file and is, to the best of our knowledge, commonly used in GIS.